# Synaptic plasticity through activation of GluA3-containing AMPA-receptors

Maria C Renner[†], Eva HH Albers[†], Nicolas Gutierrez-Castellanos, Niels R Reinders, Aile N van Huijstee, Hui Xiong, Tessa R Lodder, Helmut W Kessels*

Synaptic Plasticity and Behavior Group, The Netherlands Institute for Neuroscience, Royal Netherlands Academy of Arts and Sciences, Amsterdam, The Netherlands

**Abstract** Excitatory synaptic transmission is mediated by AMPA-type glutamate receptors (AMPARs). In CA1 pyramidal neurons of the hippocampus two types of AMPARs predominate: those that contain subunits GluA1 and GluA2 (GluA1/2), and those that contain GluA2 and GluA3 (GluA2/3). Whereas subunits GluA1 and GluA2 have been extensively studied, the contribution of GluA3 to synapse physiology has remained unclear. Here we show in mice that GluA2/3s are in a low-conductance state under basal conditions, and although present at synapses they contribute little to synaptic currents. When intracellular cyclic AMP (cAMP) levels rise, GluA2/3 channels shift to a high-conductance state, leading to synaptic potentiation. This cAMP-driven synaptic potentiation requires the activation of both protein kinase A (PKA) and the GTPase Ras, and is induced upon the activation of β-adrenergic receptors. Together, these experiments reveal a novel type of plasticity at CA1 hippocampal synapses that is expressed by the activation of GluA3-containing AMPARs.
DOI: https://doi.org/10.7554/eLife.25462.001

*For correspondence:
h.kessels@nin.knaw.nl

[†]These authors contributed equally to this work

Competing interests: The authors declare that no competing interests exist.

## Introduction

AMPA-type glutamate receptors (AMPARs) are responsible for fast excitatory synaptic transmission in the brain. AMPAR channels are formed through the assembly of four AMPAR subunits. In excitatory neurons of the mature hippocampus, the majority of AMPARs consist of subunits GluA1 and GluA2 (GluA1/2 heteromers) or GluA2 and GluA3 (GluA2/3 heteromers) (*Wenthold et al., 1996*). Each subunit has its individual properties based in part on the structure of the cytoplasmic C-terminal tail (C-tail). As an integral part of heteromeric AMPARs, the GluA2 subunit allows interactions with cytoplasmic proteins that control AMPAR transport to the cell surface and the endocytosis of AMPARs upon the induction of long-term depression of synapse strength (LTD) (*Anggono and Huganir, 2012*). The GluA1 subunit has a unique C-tail that largely excludes GluA1-containing AMPARs from entry into synapses (*Shi et al., 2001*). Upon the induction of long-term potentiation (LTP) or upon learning GluA1-containing AMPARs traffic into synapses, whereas a selective blockade of GluA1-trafficking prevent the expression of LTP and impairs memory formation (*Kessels and Malinow, 2009*; *Mitsushima et al., 2011*; *Rumpel et al., 2005*). In line with this, LTP and the formation of fear memories are severely impaired in GluA1-deficient mice (*Humeau et al., 2007*). The GluA1 C-tail contains several unique phosphorylation sites by which the trafficking of GluA1 to synapses can be regulated. For instance, the C-tail of GluA1 can be phosphorylated by protein kinase A (PKA) upon a rise in the levels of intracellular cyclic AMP (cAMP), which promotes GluA1-trafficking, lowers the threshold for LTP and facilitates the formation of memories (*Crombag et al., 2008*; *Esteban et al., 2003*; *Hu et al., 2007*; *Qian et al., 2012*). A recent study extended the AMPAR subunit-specific plasticity rules by showing that the N-terminal domain (NTD) of AMPAR subunits controls the anchoring of AMPARs at synapses, with a more stable anchoring of by the GluA2 NTD compared with the GluA1 NTD (*Watson et al., 2017*).

Whereas the roles of GluA1 and GluA2 in hippocampal synaptic plasticity are well established, the significance of GluA3 has remained enigmatic. Although CA1 neurons express substantial amounts of GluA3 (*Kessels et al., 2009*), GluA3-containing AMPARs contribute little to synaptic and extra-synaptic AMPAR currents (*Lu et al., 2009*; *Meng et al., 2003*). In hippocampal neurons that lack GluA3 LTP and LTD are intact (*Meng et al., 2003*; *Humeau et al., 2007*; *Reinders et al., 2016*) and the capacity of GluA3-deficient mice to acquire memories is comparable to wild-type congenics (*Adamczyk et al., 2012*; *Humeau et al., 2007*), inferring that GluA3-containing AMPARs do not play a prominent role in plasticity mechanisms that underlie memory formation. When recombinant GluA2/3s are expressed in CA1 neurons of hippocampal slices, they are continuously delivered to synapses independent of neuronal activity (*Shi et al., 2001*). GluA2/3s that are expressed in neurons of the cortex become enriched at synapses when these neurons are deprived of experience-dependent input, implying a role for GluA2/3s in the homeostatic scaling of synaptic strength (*Makino and Malinow, 2011*). Interestingly, the presence of GluA3 is required for amyloid-β, the prime suspect to cause Alzheimer's disease, to mediate synaptic and memory deficits (*Reinders et al., 2016*).

In this study we show that GluA3-containing AMPARs are present at CA1 synapses in hippocampal slices, but transmit no or low currents upon glutamate binding under basal conditions. However, upon an increase in intracellular cAMP, for instance upon the activation of β-adrenergic receptors, GluA3-containing AMPARs become functional, leading to a novel type of synaptic potentiation.

## Results

### cAMP selectively increases currents through GluA3-containing AMPARs

To examine the contribution of GluA3 to the pool of extra-synaptic AMPARs, we recorded currents evoked by puffing AMPA onto outside-out membrane patches excised from cell bodies of CA1 pyramidal neurons in organotypic hippocampal slices prepared from wild-type, GluA3-deficient or GluA1-deficient mice (*Figure 1A,B*). Extra-synaptic AMPAR currents recorded from GluA3-deficient neurons were similar compared to those from with wild-type neurons (t-test, p>0.9; *Figure 1A*). A non-stationary noise analysis on these outside-out AMPAR currents showed that the absence of GluA3-containing AMPAR currents did not result in differences in the average number of functional AMPARs per patch, single channel conductance or open-channel probability (*Figure 1C,D*). Consistent with previous results (*Andrásfalvy et al., 2003*), GluA1-deficient outside-out patches contained relatively few functional AMPAR channels, and those that were present had a low single-channel conductance and open channel probability (*Figure 1C,D*). As a result, currents were 83% smaller in the absence of GluA1 (t-test, p<0.0001; *Figure 1A*), indicating that under basal conditions the majority of extra-synaptic AMPAR currents are derived from GluA1-containing AMPARs.

A rise in intracellular cAMP results in a potentiation of AMPAR currents (*Carroll et al., 1998*; *Chavez-Noriega and Stevens, 1992*; *Sokolova et al., 2006*). To study how cAMP modifies AMPARs, we recorded AMPA-evoked currents from outside-out patches with or without cAMP in the solution of the recording pipette, while blocking cAMP degradation by phosphodiesterases (PDEs) with IBMX (*Figure 1A*). We note that IBMX by itself did not change the amplitude of the response to AMPA puffs (*Figure 1—figure supplement 1*). In somatic membrane patches from wild-type neurons the amplitudes of the responses to AMPA puffs were 2-fold higher in the presence of cAMP (*Figure 1A*). In GluA3-deficient somatic patches cAMP failed to change AMPAR currents *Figure 1A*). In GluA1-deficient CA1 patches AMPAR currents increased 4.5-fold in the presence of cAMP (*Figure 1A*), which occurred as a function of an increase in the estimated number of functional channels, channel conductance and open-channel probability (*Figure 1C,D*). As a result, GluA3-containing AMPARs in GluA1-deficient patches activated by cAMP reached conductance levels comparable to those of GluA1-containing AMPARs in GluA3-deficient patches, the channel properties of which were not modified by cAMP (*Figure 1C,D*). These experiments suggest that cAMP selectively potentiated currents through extra-synaptic GluA3-containing AMPARs, revealing a novel type of AMPAR plasticity.

To examine whether GluA3-dependent plasticity involves trafficking of GluA3 to the cell surface, we introduced recombinant GluA3 into GluA3-deficient CA1 neurons using Sindbis virus. As a means to quantify the surface levels of recombinant GluA3, the subunit was tagged with super ecliptic phluorin (SEP), a GFP variant that is only fluorescent at neutral pH, i.e. when present in the ER, cis-

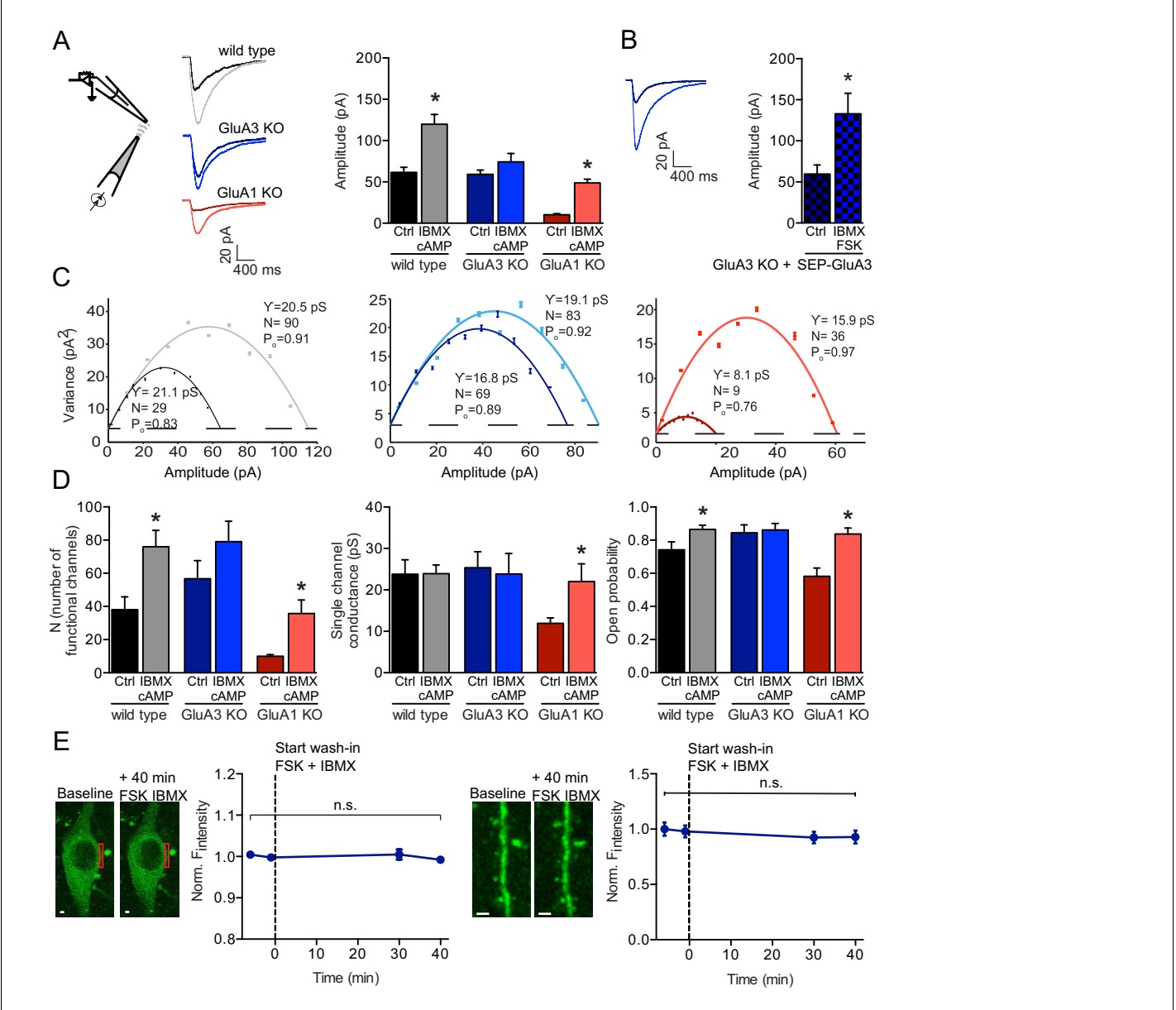

**Figure 1.** cAMP increases currents of extrasynaptic GluA3-containing AMPARs. (**A**) Experimental setup, example traces and averages of AMPAR amplitudes in response to AMPA puffs onto outside-out patches from WT CA1 cell bodies without (black; n=15) or with (gray; n=24) cAMP in pipette, GluA3-KO without (dark blue; n=20) or with (light blue; n=20) cAMP, and GluA1-KO without (dark red; n=18) or with (light red; n=10) cAMP. Wild type and GluA1-deficient neurons showed an increased amplitude upon AMPA puffs in the presence of cAMP in the recording pipette, GluA3-deficient did not (t-test, WT: p=0.0009; t-test, 3KO: p=0.9; t-test, 1KO: p<0.0001). All recordings in the presence of desensitization blockers cyclothiazide and PEPA and cAMP condition in the presence of PDE inhibitor IBMX. (**B**) Outside-out patches from SEP-GluA3 infected GluA3-KO cell bodies, before (n=7) and after (n=8) wash-in of forskolin plus IBMX. Sindbis expression of SEP-GluA3 in GluA3-KO neurons rescues GluA3-dependent plasticity (t-test, p=0.03). (**C**) Example graphs of non-stationary noise analysis for WT, GluA3-KO and GluA1-KO outside-out recordings. (**D**) Average number of functional channels, single channel conductance and open probability extracted from non-stationary noise analysis of the outside-out recordings from WT (ctrl: n=14; FSK/IBMX: n=16), GluA3-KO (ctrl: n=10; FSK/IBMX: n=11), or GluA1-KO (ctrl: n=8; FSK/IBMX: n=7) CA1 neurons (see methods). Upon forskolin and IBMX GluA1-deficient neurons show an increased number of functional channels (t-test, p=0.0009), single channel conductance (t-test, p=0.04) and open probability (t-test, p=0.002). (**E**) Example images and average fluorescence intensity of GluA3-KO cell bodies (left; n=10) and dendrites (right; n=12) infected with SEP-GluA3 visualized with 2-photon laser scanning microscopy before and after wash-in of forskolin plus IBMX shows no change in SEP levels (dendrite: t-test, p=0.4; cell bodies: t-test, p=0. 2). Scale bars indicate 5 μm. Error bars indicate SEM, * indicates p<0.05.

DOI: https://doi.org/10.7554/eLife.25462.002

The following figure supplements are available for figure 1:

*Figure 1 continued on next page*

*Figure 1 continued*

**Figure supplement 1.** Blocking phosphodiesterase activity is not sufficient to induce significant AMPAR potentiation.
DOI: https://doi.org/10.7554/eLife.25462.003

**Figure supplement 2.** SEP-GluA3 fluorescence is pH-sensitive.
DOI: https://doi.org/10.7554/eLife.25462.004

Golgi or at the cell surface, but not at acidic pH in late endosomes or lysosomes. We visualized the pH-sensitivity of SEP-GluA3 by acutely washing in ACSF buffered at pH 5 (*Figure 1—figure supplement 2*). Whereas SEP-GluA3 expression in GluA3-deficient neurons did not alter outside-out AMPAR currents, washing in both the adenylyl cyclase activator forskolin and PDE inhibitor IBMX rescued the cAMP-driven potentiation of AMPAR currents (*Figure 1B*), indicating that recombinant GluA3 responds similarly to a rise in cAMP as endogenous GluA3. However, the SEP fluorescence levels remained unchanged at the surface area of both dendrites and cell bodies upon the forskolin/IBMX wash-in (*Figure 1E*), suggesting that cAMP-driven GluA3-plasticity does not result from trafficking of GluA3-containing AMPARs from endocytic compartments to the cell surface.

## cAMP increases the open-channel probability of GluA3

We examined whether cAMP-signaling can activate GluA2/3-channel function by performing single-channel recordings under cell-attached configuration from soma of CA1 pyramidal GluA1- or GluA3-deficient neurons from organotypic hippocampal slices, while adding AMPA to the recording pipette. AMPARs can have different conductance states depending on the number of AMPAR subunits that effectively bind glutamate (*Gebhardt and Cull-Candy, 2006*; *Rosenmund et al., 1998*). Three different open states could be distinguished in our single-channel AMPAR recordings (*Figure 2A,B*). Single-channel currents of AMPARs recorded from GluA3-deficient neurons did not change in the presence of forskolin (*Figure 2A*), indicating that the channel properties of GluA1-containing AMPARs remained unchanged upon a rise in cAMP. In contrast, GluA2/3s at the surface of GluA1-deficient neurons had a low open-channel probability under basal conditions and remained largely stuck in open state 1 (*Figure 2B*). In the presence of forskolin the open-channel probability was on average six-fold higher and GluA3-containing AMPARs more often reached open states 2 and 3. Their average conductance increased without modifying the conductance level per open state. These experiments indicate that GluA3-containing AMPARs are largely electrically quiet under basal conditions, but become functional upon a rise in cAMP by increasing their capacity for glutamate-gated channel opening.

## A rise in cAMP activates GluA3-containing AMPARs at synapses

To investigate whether the cAMP-driven activation of GluA3-dependent plasticity occurs at synapses, we performed whole-cell patch clamp recordings on CA1 neurons from organotypic hippocampal slices in the presence or absence of forskolin. Forskolin incubation significantly increased AMPA/NMDA ratios evoked by stimulating Schaffer collateral inputs onto synapses of wild-type CA1 neurons (*Figure 3A*). Consistent with the notion that the potentiation effect of forskolin is predominantly postsynaptic (*Sokolova et al., 2006*), forskolin did not affect presynaptic glutamate release, since paired pulse ratios did not change (ANOVA: WT: p=0.8, KO: p=0.9; *Figure 3B*) and the quantal content of evoked glutamate release determined by a variance analysis (*Figure 3—figure supplement 1*) did not increase. The variance analysis also revealed that AMPA/NMDA ratios increased as a function of increased AMPAR currents and not decreased NMDAR currents (*Figure 3—figure supplement 1*). At Sc-CA1 synapses of GluA3-deficient neurons however, forskolin did not change AMPA/NMDA ratios (*Figure 3A*), indicating that a rise in cAMP triggers potentiation of postsynaptic GluA3-containing AMPARs.

In miniature EPSC (mEPSC) recordings, the forskolin-driven postsynaptic potentiation was reflected as an increase in the frequency and amplitude of mEPSC events (*Figure 3C*). In GluA3-deficient neurons, forskolin did not significantly change the average mEPSC frequency or amplitude (*Figure 3C*). Re-introducing GluA3 in GluA3-deficient CA1 neurons by acute viral expression of GFP-GluA3 and allowing it to be expressed for 24 hr partially rescued the forskolin-driven synaptic potentiation (*Figure 3C*). GluA3-dependent synaptic potentiation was also induced by directly infusing

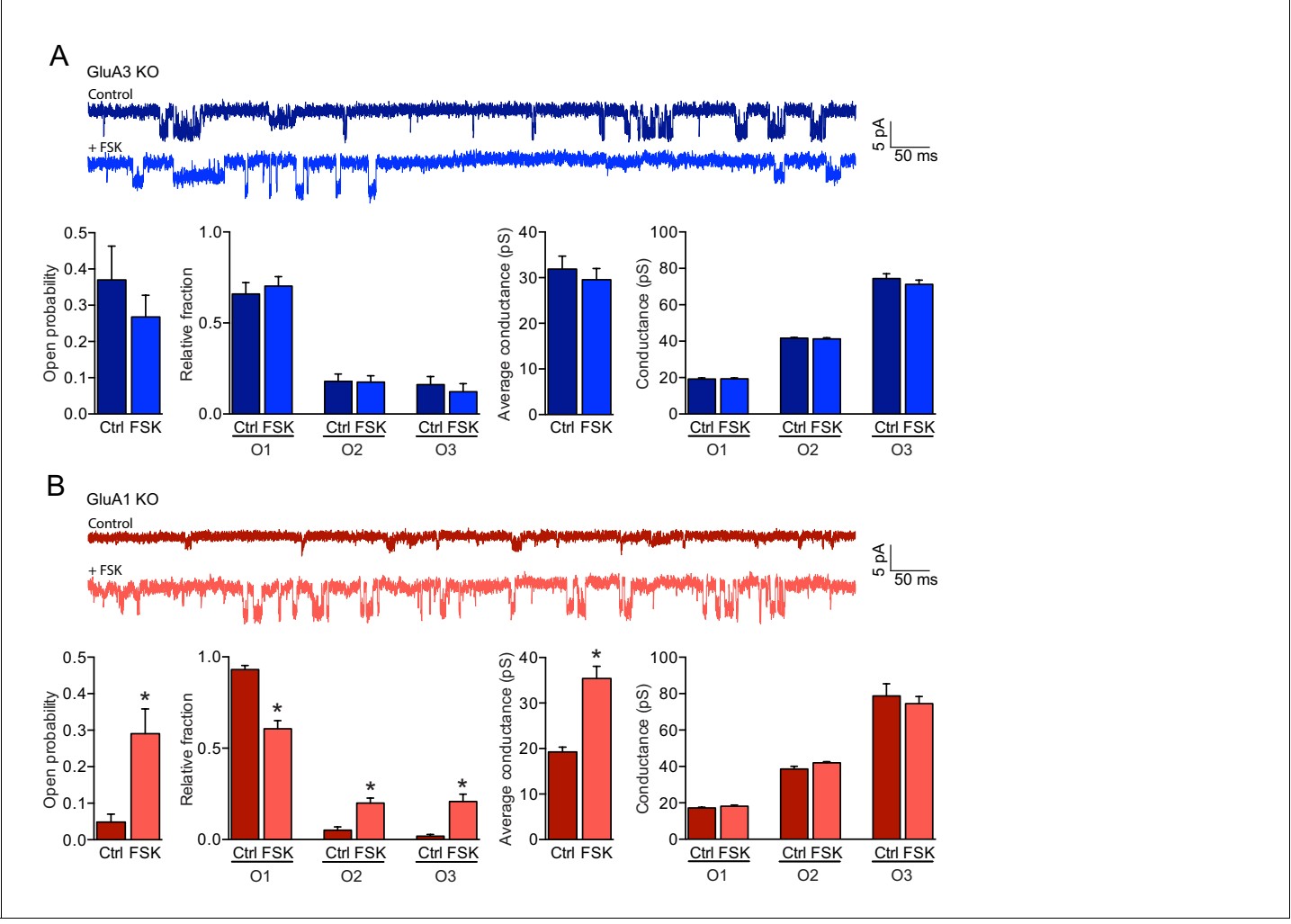

**Figure 2.** cAMP increases the open-channel probability of GluA3-containing AMPARs. (A, B) Single channel recordings under cell-attached configuration showing example traces, open probability, fraction of time spent in each open state, average conductance and conductances of each open state. (A) GluA1-containing AMPARs on GluA3-KO neurons (blue) did not respond to forskolin (ctrl: n=15, FSK: n=14) (open-channel probability: t-test, p=0.9; single-channel conductance: t-test, p=0.5). (B) The open-probability (t-test, p=0.0001) and single-channel conductance (t-test: p<0.0001) increased of GluA3-containing AMPARs on GluA1-KO neurons (red) upon incubation with forskolin (ctrl: n=15, FSK n=14). Error bars indicate SEM, * indicates p<0.05.

DOI: https://doi.org/10.7554/eLife.25462.005

cAMP into CA1 neurons with PDEs blocked by IBMX (*Figure 3—figure supplement 2*). Neurons that are deficient in both GluA1 and GluA3 virtually lacked mEPSC events (*Figure 3D*), indicating that synaptic AMPAR currents recorded from GluA1-deficient CA1 neurons predominantly stem from GluA3-containing AMPARs. Whereas these GluA1/GluA3 double deficient neurons were insensitive to forskolin, forskolin did elevate the mEPSC frequency in GluA1-deficient neurons (*Figure 3B*). To determine whether the cAMP-driven synaptic potentiation could be explained by increased single-channel conductance, we conducted peak-scaled non-stationary fluctuation analyses from mEPSCs. The synaptic AMPAR single-channel conductance was significantly increased by forskolin only when cells express GluA3 (*Figure 3E,F*). These experiments indicate that postsynaptic currents of GluA3-containing AMPARs increase upon a rise in cAMP. A postsynaptic potentiation of GluA2/3 currents that is reflected by an increase in frequency rather than average amplitude may be explained by mEPSCs rising above the detection limit for mEPSC analysis. The distribution of mEPSC amplitudes recorded from GluA1-deficient CA1 neurons suggests that indeed a substantial

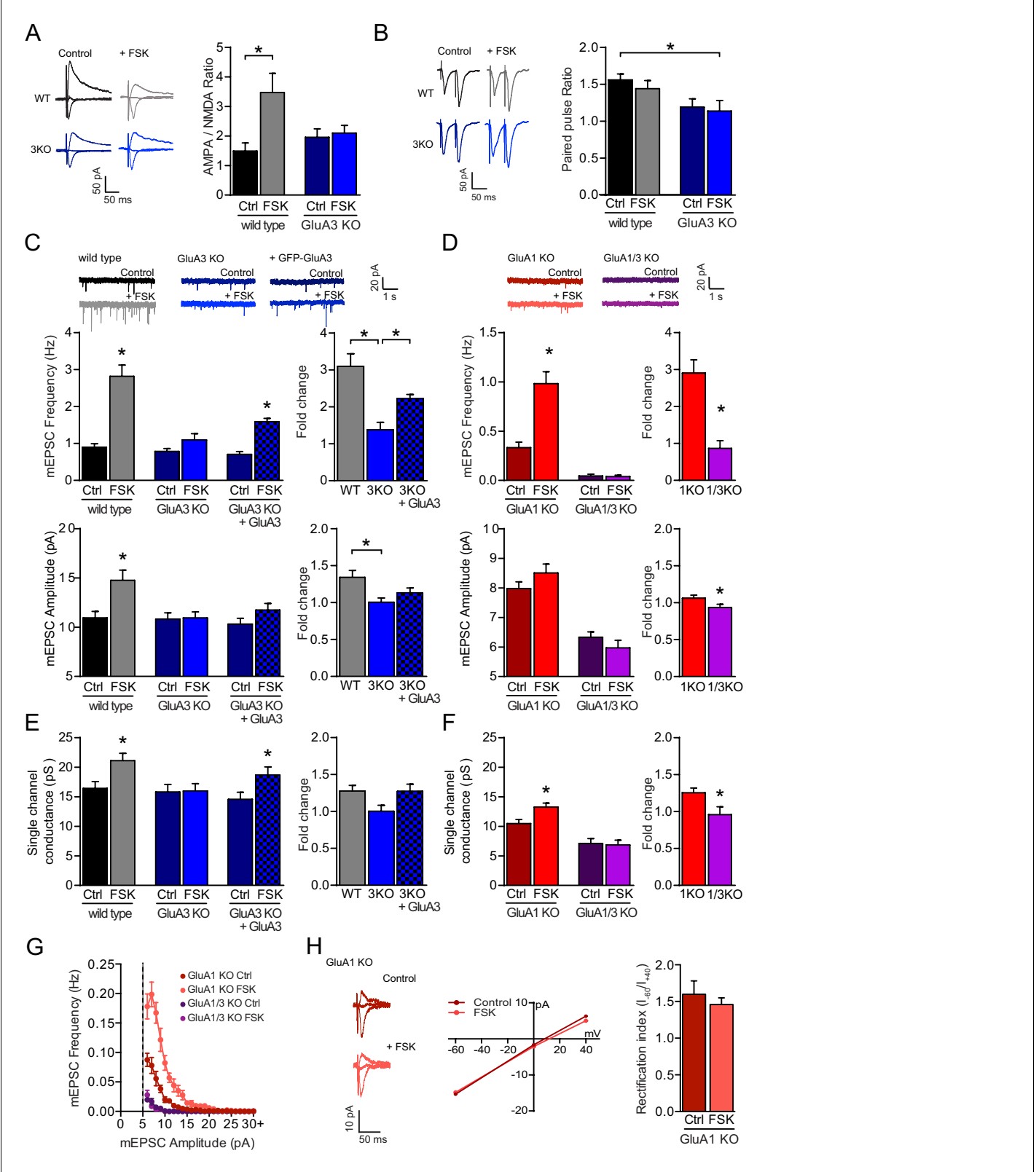

**Figure 3.** The cAMP-driven postsynaptic potentiation depends on GluA3. (**A**) Example traces and average AMPA/NMDA EPSC ratios of WT neurons with (grey; n=13) or without (black; n=9) forskolin, and GluA3-KO neurons with (blue; n=8) or without (dark blue; n=8) forskolin. In WT neurons the AMPA/NMDA ratio increased upon forskolin incubation (ANOVA, p=0.02), but not in GluA3-KO neurons (ANOVA, p>0.9). (**B**) Example traces and average paired pulsed ratio, which were not different in WT CA1 neurons with (n=8) or without (n=8) forskolin (ANOVA: p=0.9), or of GluA3-KO neurons

*Figure 3 continued on next page*

Figure 3 continued

with (n=8) or without (n=8) forskolin (ANOVA: p>0.9). (C) In WT neurons mEPSC frequencies (t-test, p<0.0001) and amplitudes (t-test, p=0.001) were higher in presence (n=16) versus absence (n=15) of forskolin. In GluA3-KO CA1 the mEPSC frequency (t-test, p=0.15) and amplitude (t-test, p=0.9) were not different in presence (n=9) or absence (n=8) of forskolin. In Sindbis-infected GluA3-KO neurons expressing GFP-GluA3 (blocked bars) the mEPSC frequency (t-test, p<0.0001), but not amplitude (t-test, p=0.1), increased in presence (n=11) versus absence (n=8) of forskolin. The forskolin-driven fold increase in mEPSC frequency (ANOVA, p<0.0001) and amplitude (ANOVA, p=0.006) was higher in WT versus GluA3-KO neurons. This fold increase in GFP-GluA3 infected versus uninfected GluA3-KO neurons was larger in mEPSC frequency (ANOVA, p=0.007), but not amplitude (ANOVA, p=0.4). (D) Forskolin increased mEPSC frequencies (t-test, p<0.0001) and amplitudes (t-test, p=0.15) of GluA1-KO neurons (ctrl: dark red; n=11; FSK: light red; n=10), but not in GluA1/3-KO neurons (ctrl: dark purple; n=11; FSK: light purple; n=8). The forskolin-driven fold increase in mEPSC frequency (t-test, p<0.0001) and amplitude (t-test, p=0.03) was larger in GluA1-KO versus GluA1/3-KO. (E) Non-stationary noise analysis of scaled mEPSCs revealed single-channel conductance of WT neurons with (n=10) or without (n=13) forskolin (t-test, p=0.009), GluA3-KO neurons with (n=8) or without (n=8) forskolin (t-test, p=0.9), and GFP-GluA3-expressing GluA3-KO neurons with (n=11) or without (n=8) forskolin (t-test, p=0.03), and forskolin-driven fold increase in single-channel conductance (WT vs GluA3-KO: ANOVA, p=0.07; GluA3-KO uninf. vs GFP-GluA3-inf: ANOVA, p=0.07). (F) Non-stationary noise analysis of scaled mEPSCs revealed single-channel conductance of GluA1-KO neurons with (n=8) or without (n=12) forskolin (t-test, p=0.01) and GluA1/3-KO neurons with (n=6) or without (n=11) forskolin (t-test, p=0.9), and forskolin-driven fold increase in single-channel conductance (t-test, p=0.02). (G) mEPSC distribution per 1 pA binned amplitude of GluA1-KO neurons without (dark red; n=11) or with (light red; n=10) forskolin and GluA1/3-KO neurons without (dark purple; n=11) or with (light purple; n=8) forskolin. (H) GluA3 exists as GluA2/3 heteromeric AMPARs. Example traces with corresponding I-V curve and average rectification indices (($I_{-60mV} - I_{0mV}$) / ($I_{+40mV} - I_{0mV}$)) were determined in GluA1-KO organotypic slices of CA1 neurons without (n=6) or with (n=6) forskolin treatment (t-test, p=0.7). Error bars indicate SEM, * indicates p<0.05.

DOI: https://doi.org/10.7554/eLife.25462.006

The following figure supplements are available for figure 3:

**Figure supplement 1.** Forskolin increases postsynaptic AMPAR currents.
DOI: https://doi.org/10.7554/eLife.25462.007

**Figure supplement 2.** Postsynaptically applied cAMP causes a GluA3-dependent synaptic potentiation.
DOI: https://doi.org/10.7554/eLife.25462.008

**Figure supplement 3.** Sindbis viral GFP-GluA3 expression predominantly traffic into CA1 synapses in configuration of GluA2/3 heteromers.
DOI: https://doi.org/10.7554/eLife.25462.009

proportion of mEPSC events fell below the 5 pA detection limit, and this proportion appeared smaller in the presence of forskolin (*Figure 3G*).

GluA3-containing AMPARs obligatorily exist as heteromers (*Coleman et al., 2016*), likely because structural constraints at their N-terminal domain prevent GluA3s from efficiently forming homomers (*Herguedas et al., 2016*; *Sukumaran et al., 2011*). To test whether the forskolin-driven GluA3-dependent plasticity was mediated by GluA2/3 heteromers, we measured the AMPAR rectification indices in GluA1-deficient neurons. Unlike GluA2-containing AMPARs, GluA2-lacking AMPARs are rectifying: they conduct more easily at negative potentials. GluA1-deficient neurons displayed non-rectifying currents both in the absence and presence of forskolin (*Figure 3H*). Also when GFP-GluA3 was overexpressed in GluA3-deficient neurons, AMPAR currents remained non-rectifying in the presence of forskolin (*Figure 3—figure supplement 3*). These data indicate that GluA3-dependent plasticity at synapses predominantly stems from GluA2/3 heteromers.

## Activation of GluA2/3s does not involve AMPAR trafficking

To examine whether cAMP-driven AMPAR-plasticity is accompanied with changes in AMPAR synaptic trafficking or lateral mobility, we performed time-lapse 2-photon imaging of CA1 neurons in wild-type organotypic slices infected with Sindbis virus to acutely express SEP-GluA3 or SEP-GluA1 together with cytoplasmic marker tdTomato. 24 hr after viral infection we assessed the fluorescence recovery after photobleaching (FRAP) of single spines (*Figure 4A*). The FRAP protocol did not affect the size of the photobleached spines nor the fluorescence signal in neighboring spines (*Figure 4—figure supplement 1*). After bleaching, the SEP-GluA3 and SEP-GluA1 fluorescence intensity recovered at a similar pace irrespective of forskolin/IBMX being present (*Figure 4B,C*). These data indicate that GluA3 and GluA1 show a similar lateral mobility at the surface of spines, which did not change upon a rise in cAMP.

Since AMPARs become immobilized when incorporated into synapses, the level of FRAP is indicative for the fraction of SEP-GluA3 or SEP-GluA1 located at synapses. As shown previously (*Makino and Malinow, 2009*), SEP-GluA1 fluorescence recovered to ~100% (*Figure 4B*), indicating that GluA1-containing AMPARs are largely excluded from entry into synapses under basal

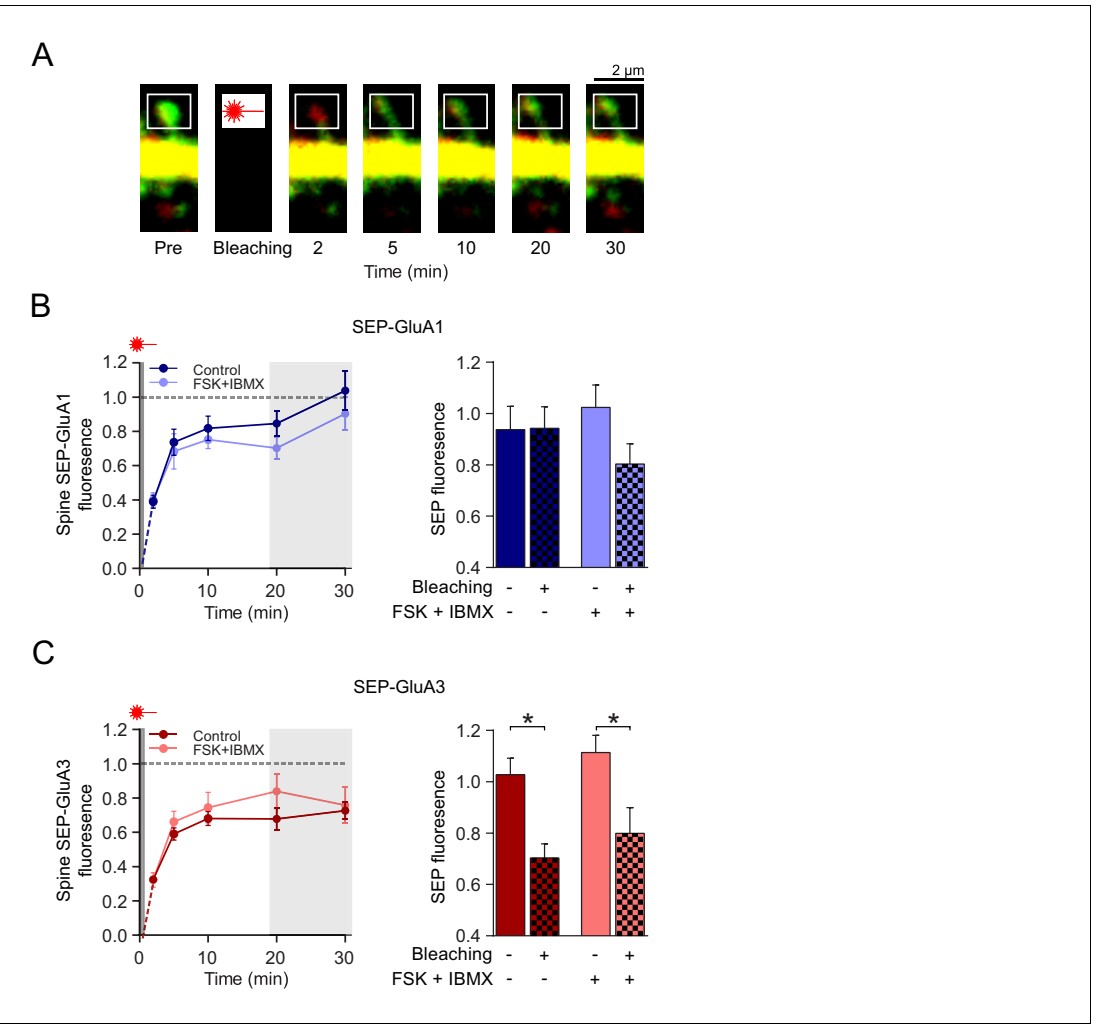

**Figure 4.** cAMP does not affect GluA2/3 mobility or trafficking to synapses. (**A**) Example image of a FRAP experiment: a spine from a CA1 neuron transfected with SEP-GluA3 + tdTomato obtained with two-photon microscopy immediately before, and 2, 5, 10, 20 and 30 min after photobleaching of the spine. (**B**) Left: Quantification of FRAP dynamics of spines transfected with SEP-GluA1 + tdTomato in the absence (dark blue; n=8) and the presence (light blue; n=6) of forskolin and IBMX. Right: Average fluorescence at time-points 20 min and 30 min after photobleaching of bleached spines (Ctrl: n=7, FSK/IBMX: n=8) in comparison to non-bleached neighboring spines (Ctrl: n=6, FSK/IBMX: n=6) in absence (ANOVA, p>0.9) or presence (ANOVA, p=0.3) of forskolin (see *Figure 4—figure supplement 1*). (**C**) Left: Quantification of FRAP dynamics of spines transfected with SEP-GluA3 + tdTomato in the absence (dark red; n=7) and the presence (light red; n=8) of forskolin and IBMX. Right: Average fluorescence at time-points 20 min and 30 min after photobleaching of bleached spines in comparison to unbleached neighboring spines (Ctrl: n=5, FSK/IBMX: n=5) in the absence (ANOVA, p=0.04) or presence (ANOVA, p=0.04) of forskolin (see *Figure 4—figure supplement 1*). Error bars indicate SEM, * indicates p<0.05.

DOI: https://doi.org/10.7554/eLife.25462.010

The following figure supplement is available for figure 4:

**Figure supplement 1.** Controls in FRAP experiment.
DOI: https://doi.org/10.7554/eLife.25462.011

conditions. In the presence of forskolin/IBMX the recovery of SEP-GluA1 was lower (~80%) although this did not reach significance (*Figure 4B*). SEP-GluA3 recovered to ~75% both in the absence and presence of forskolin/IBMX (*Figure 4C*), demonstrating that GluA2/3s are constitutively inserted into synapses independent of cAMP levels. These experiments are consistent with the model that

synaptic GluA3-plasticity is expressed by an increase in channel conductance of GluA3-containing AMPARs already present at synapses.

## A cAMP-driven activation of GluA2/3s requires PKA and Ras activity

We next set out to identify the cAMP-dependent mediator that triggers GluA3-plasticity. A rise in cAMP can cause postsynaptic potentiation by triggering GluA1 phosphorylation via PKA activation (*Joiner et al., 2010*; *Man et al., 2007*). In line with this, adding the selective PKA activator N002 to the recording pipette significantly increased mEPSC frequency and amplitude in wild-type CA1 neurons (*Figure 5A*), and not in GluA1-deficient CA1 neurons (*Figure 5B*). Pre-incubation of wild-type CA1 neurons with PKA-inhibitors PKI (*Figure 5C*) or KT 5720 (*Figure 5—figure supplement 1*) failed to prevent the increase in mEPSC frequency. In GluA1-deficient CA1 neurons PKI also did not significantly block the forskolin-driven increase in mEPSC frequency (*Figure 5D*). These data indicate that PKA activation can trigger GluA1-dependent synaptic potentiation, but is not sufficient to promote GluA2/3-plasticity at synapses. We further excluded the involvement of the cAMP-dependent activation of HCN channels in GluA2/3-plasticity, since these channels were blocked by intracellular cesium during whole-cell recordings. We also found no evidence supporting a direct activation of GluA2/3 receptor complexes by cAMP: applying cAMP directly onto inside-out patches obtained from cell bodies of GluA1-deficient CA1 neurons did not change AMPAR conductance or open-channel probability (*Figure 5—figure supplement 2*).

A rise in cAMP can activate the small GTPases Rap1 and Ras (*Li et al., 2016*). Rap1 is activated by the Epac family of the cAMP-regulated guanidyl exchange factors (GEFs) (*Gloerich and Bos, 2010*). Adding the Epac activator 8-CPT-2Me-cAMP (8CPT) to the recording pipette did not induce a change in mEPSC amplitude or frequency in CA1 neurons (*Figure 5E*), and a drug (ESI05) that inhibits Epac was unable to block the effect of forskolin (*Figure 5F*). In line with previous studies (*Gelinas et al., 2008*; *Ster et al., 2009*; *Zhu et al., 2002*), these data show that Epac/Rap1 signaling does not induce synaptic potentiation in CA1 neurons. Ras activation can be triggered by cAMP-dependent GEFs Ras-GRF1 and/or PDZ-GEF1 (*Ambrosini et al., 2000*; *Fasano et al., 2009*; *Li et al., 2016*) and leads to a synaptic potentiation in CA1 neurons (*Zhu et al., 2002*). For Ras signaling to occur at the cell membrane, a farnesyl group needs to be attached to the pre-Ras protein by farnesyltransferase (FT). The FT-inhibitor Salirasib partially blocked the forskolin-mediated increase in mEPSC frequency (*Figure 5—figure supplement 3*), indicating that GluA2/3-plasticity involves a mediator that depends on FT activity. To directly test whether Ras mediates the cAMP-driven synaptic potentiation, CA1 neurons in acutely isolated brain slices were internally perfused with a Ras-specific IgG antibody, which binds the switch II region of H-, K-, and N-Ras, thereby inhibiting its conformational activation (*Cardinale et al., 1998*), or with a control IgG. The infusion of anti-Ras or control IgG did not affect basal synaptic transmission in both wild-type and GluA1-deficient neurons (*Figure 6—figure supplement 1*). Upon the wash-in of forskolin, the mEPSC frequency gradually increased irrespective of the anti-Ras or control IgG being present (*Figure 6A,B*, left), indicating that a blockade of Ras is not sufficient to prevent the cAMP-driven potentiation of GluA2/3 currents. A previous study showed that in neurons the cAMP-driven activation of Ras is in part dependent on PKA (*Ambrosini et al., 2000*). To test whether PKA and Ras cooperate to mediate cAMP-driven synaptic potentiation, we repeated this experiment in the presence PKA-inhibitor PKI (*Figure 6A,B*, right and *Figure 6—figure supplement 2*). When both PKA and Ras were blocked in wild-type CA1 neurons, the mEPSC frequency did not increase upon forskolin wash-in, although the forskolin-mediated fold increase in mEPSC frequency was not significantly different between Ras and ctrl IgG (*Figure 6A*). In GluA1-deficient neurons, in which the GluA2/3 currents are isolated, the blockade of both Ras and PKA did fully prevent the forskolin-driven fold increase in mEPSC frequency (*Figure 6B*). These experiments indicate that the cAMP-driven signaling pathway that triggers activation of GluA2/3-plasticity requires the activation of both PKA and Ras.

## β-adrenergic signaling triggers the activation of GluA2/3-plasticity

A rise in intracellular cAMP in neurons can be achieved upon the release of norepinephrine (NE) through the activation of β-adrenergic receptors (β-ARs) (*Seeds and Gilman, 1971*) and β-ARs activation by selective agonist isoproterenol induces both PKA and Ras signaling (*Enserink et al., 2002*). Isoproterenol by itself is known to generate only a weak increase in intracellular cAMP and

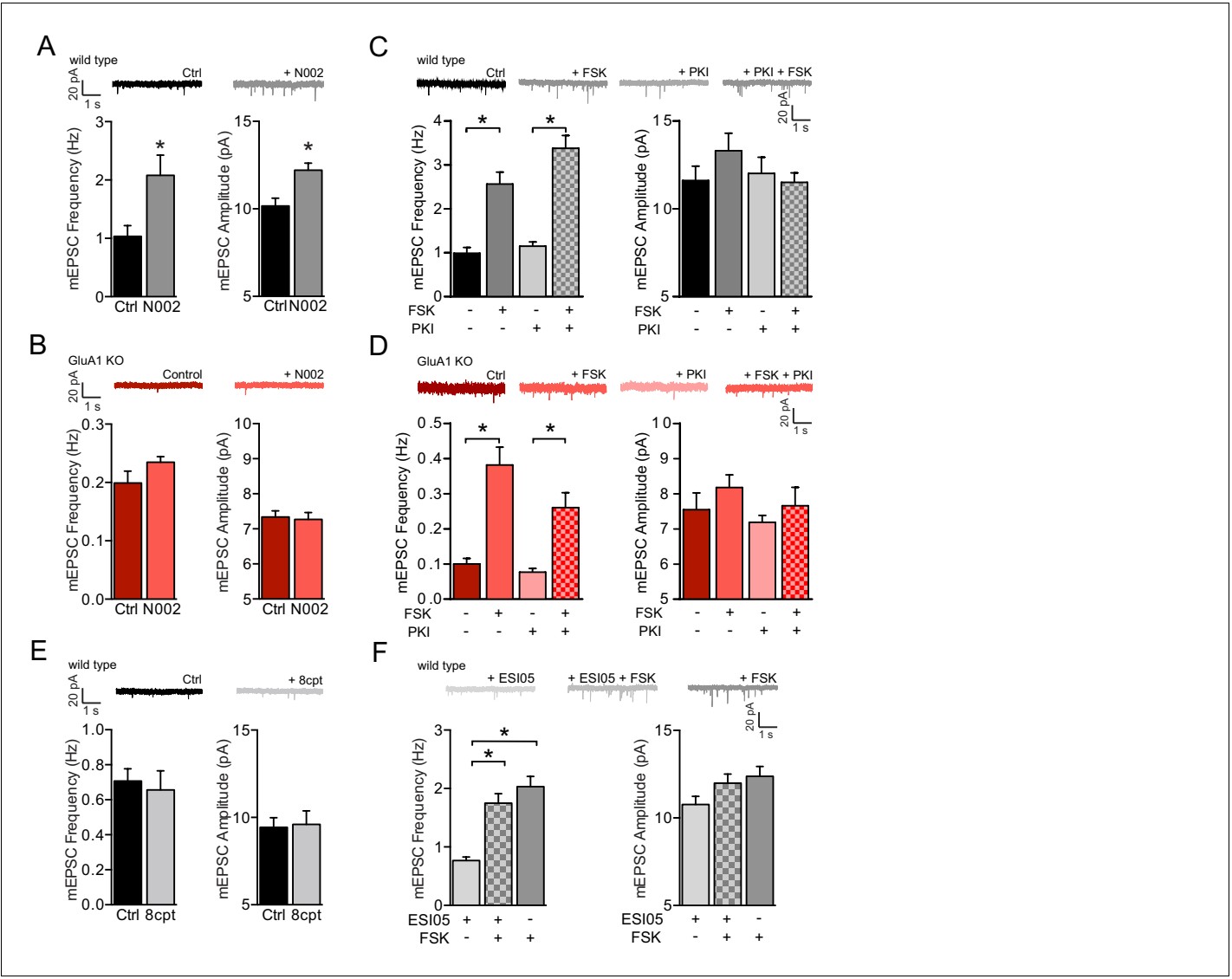

**Figure 5.** PKA and Epac activation are not sufficient to activate GluA2/3-plasticity. (**A**) Example traces, mEPSC recordings of WT neurons with (n=12) or without (n=13) PKA agonist N002 in the recording pipette. N002 increased mEPSC frequency (t-test, p=0.004) and mEPSC amplitude (t-test, p=0.007). (**B**) Example traces, mEPSC recordings from GluA1-KO neurons with (n=13) or without (n=15) N002 in the recording pipette. N002 did not increase mEPSC frequency (t-test, p=0.06) and mEPSC amplitude (t-test, p=0.8) when GluA1 is not expressed. (**C**) Example traces, mEPSC frequencies and mEPSC amplitudes of untreated CA1 neurons (n=8), neurons incubated with forskolin (n=9), incubated with PKA inhibitor PKI (n=7), or preincubated with PKI prior to forskolin (n=10). PKI did not prevent the forskolin-driven increase in mEPSC frequency (ANOVA, ctrl vs FSK: p<0.0001, PKI vs FSK/PKI: p<0.0001). (**D**) Same as for (**B**) for GluA1-KO CA1 neurons either untreated (n=4), incubated with forskolin (n=5), incubated with PKI (n=5), or preincubated with PKI prior to forskolin (n=5) (ANOVA, ctrl vs FSK: p<0.0001, PKI vs FSK/PKI: p=0.0001). (**E**) mEPSC recordings from wildtype CA1 neurons with control intracellular solution (n=7) or with Epac activator 8-CPT-2Me-cAMP in the recording pipette (n=7). 8-CPT-2Me-cAMP did not increase average mEPSC frequency (t-test, p=0.5) or amplitude (t-test, p=0.9) (**F**) WT CA1 neurons incubated with Epac inhibitor ESI-05 (n=12), forskolin (n=10) or preincubated with ESI-05 prior to forskolin (n=15). ESI-05 did not prevent the forskolin-driven synaptic potentiation (ANOVA, ESI-05 vs ESI-05/FSK: p<0.0001; ESI-05 vs FSK: p<0.0001; ESI-05/FSK vs FSK: p=0.4). Error bars indicate SEM, * indicates p<0.05.
DOI: https://doi.org/10.7554/eLife.25462.012

The following figure supplements are available for figure 5:

**Figure supplement 1.** PKA-inhibitor KT5720 prevents the forskolin driven increase in mEPSC amplitude, but not frequency.
DOI: https://doi.org/10.7554/eLife.25462.013

**Figure supplement 2.** Cyclic AMP does not directly activate GluA2/3-receptor complexes.
DOI: https://doi.org/10.7554/eLife.25462.014

**Figure supplement 3.** Farnesyltransferase antagonist Salirasib, a non-selective Ras inhibitor, partly blocks cAMP-driven synaptic potentiation.
DOI: https://doi.org/10.7554/eLife.25462.015

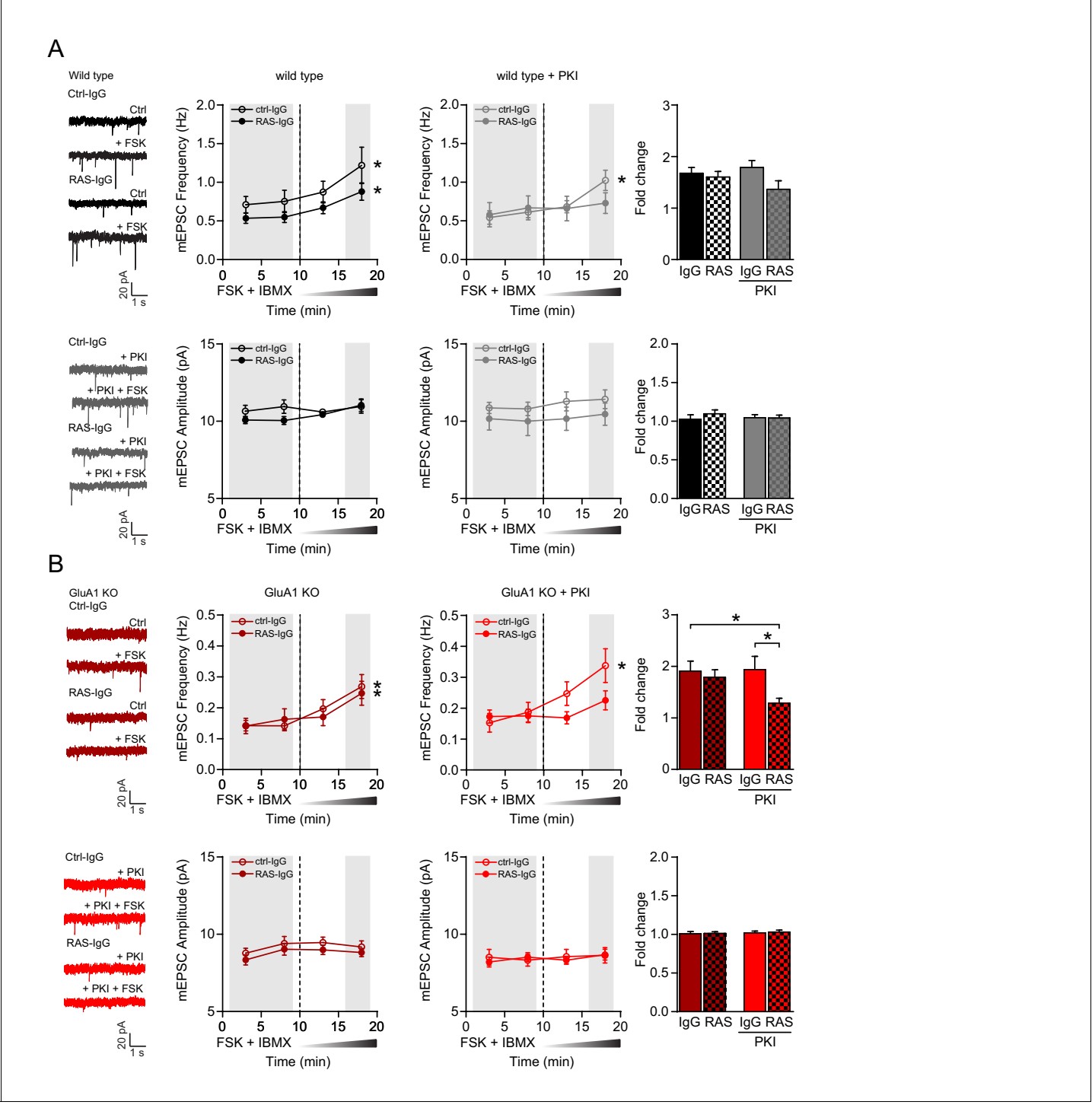

**Figure 6.** cAMP activates GluA2/3-plasticity through a PKA- and Ras-dependent signaling pathway. (A–B) Miniature EPSC recordings with anti-Ras IgG or control IgG in recording pipette from CA1 neurons in brain slices acutely isolated from mature wild type and GluA1-KO mice. After ten minutes baseline recording to allow antibody to perfuse into the cell, forskolin and IBMX were washed in the extracellular solution. (A) Upon the wash-in of forskolin, WT neurons showed an increase in mEPSC frequency in the presence of control IgG (paired t-test, p<0.001) and anti-Ras IgG (paired t-test, p<0.001), but not in mEPSC amplitude (paired t-test, ctrl-IgG: p=0.7; RAS-IgG: p=0.055). Stars indicate a significant increase at 15–20 min compared to the 10 min baseline. In the presence of PKI, the wash-in of forskolin led to a significant increase in frequency in neurons perfused with control IgG (paired t-test, p=0.007), but not with Ras IgG (paired t-test: p=0.07). Comparisons between the fold changes did not yield significant results (ANOVA, p=0.12; IgG: n=7; Ras: n=9; IgG/PKI: n=10; Ras/PKI: n=9). (B) In GluA1-deficient neurons, the wash in of forskolin also increased mEPSC frequency regardless of the antibody (paired t-test, ctrl-IgG: p<0.001; Ras-IgG: p<0.0001). mEPSC amplitudes were not affected by forskolin (paired t-test, ctrl-

*Figure 6 continued on next page*

*Figure 6 continued*

IgG: p=0.7; Ras-IgG: p=0.3). In the presence of PKI neurons perfused with control IgG show a significant increase in frequency upon the wash in of forskolin (paired t-test, p=0.006). In neurons with Ras-IgG this potentiation was absent (paired t-test: p=0.15). The fold increase in mEPSC frequency is blocked by the anti-Ras antibody in the presence of PKI in GluA1-KO neurons (ANOVA, IgG/PKI vs Ras/PKI: p=0.03; IgG vs Ras/PKI: p=0.04; IgG: n=11; Ras-IgG: n=12; IgG/PKI: n=10; Ras-IgG/PKI: n=14). Error bars indicate SEM.

DOI: https://doi.org/10.7554/eLife.25462.016

The following figure supplements are available for figure 6:

**Figure supplement 1.** The infusion of anti-Ras IgG or control IgG did not affect basal mEPSC frequency or amplitude.

DOI: https://doi.org/10.7554/eLife.25462.017

**Figure supplement 2.** Frequency distribution of mEPSC events.

DOI: https://doi.org/10.7554/eLife.25462.018

inefficiently triggers downstream signaling due to a negative feedback loop activating PDEs (*Chay et al., 2016*; *Bruss et al., 2008*; *Houslay and Baillie, 2005*). To assess whether isoproterenol can induce GluA2/3-plasticity, we performed a mEPSC analysis on CA1 excitatory neurons of the dorsal hippocampus in brain slices acutely isolated from mature mice. When these slices were incubated with isoproterenol we observed an increased frequency, but not amplitude, of mEPSC events in CA1 neurons of both wild-type and GluA1-deficient mice, provided that PDE activity was inhibited with IBMX (*Figure 7*). Isoproterenol did not trigger synaptic potentiation in CA1 neurons of GluA3-

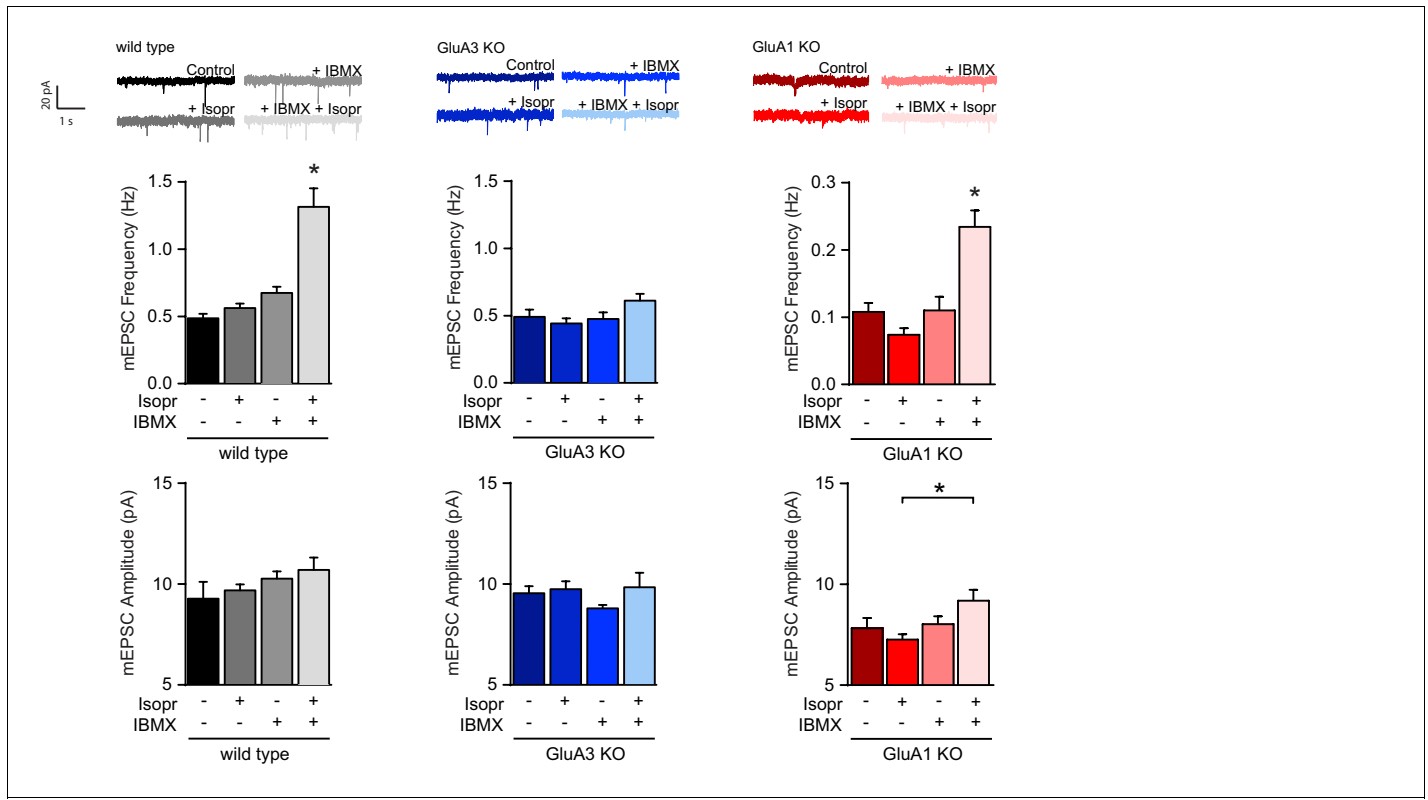

**Figure 7.** β–adrenergic activation induces GluA2/3–plasticity. Brain slices were acutely isolated from mature WT and littermate GluA3-KO mice, or GluA1-KO mice. Example traces, average mEPSC frequencies and amplitudes from CA1 neurons incubated with no drugs, b-AR agonist isoproterenol, PDE blocker IBMX, or isoproterenol plus IBMX (WT: ctrl n=13, Iso n=11, IBMX n=7, Iso + IBMX n=9; GluA3-KO: ctrl n=15, Iso n=7, IBMX n=7, Iso+IBMX n=4; GluA1-KO: ctrl n=5, Iso n=4, IBMX n=8, Iso+IBMX n=6). Isoproterenol in presence of IBMX increases mEPSC frequency in WT (ANOVA, ctrl vs Isopr: p=0.8; IBMX vs Isopr/IBMX: p<0.0001), and GluA1-KO (ANOVA, ctrl vs Isopr: p=0.4; IBMX vs Isopr/IBMX: p=0.001), but not in GluA3-KO (ANOVA, ctrl vs Isopr: p=0.9; IBMX vs Isopr/IBMX: p=0.5). Isoproterenol/IBMX did not affect average mEPSC amplitude in WT (ANOVA, p=0.4), in GluA3-KO (ANOVA, p=0.4), or in GluA1-KO neurons (ANOVA, Isopr vs isopr/IBMX p=0.045). Error bars indicate SEM, * indicates p<0.05.

DOI: https://doi.org/10.7554/eLife.25462.019

deficient brain slices (*Figure 7*), indicating that β-AR activation can evoke GluA3-dependent plasticity upon a robust increase in cAMP levels.

To examine whether GluA3-dependent plasticity can be induced in vivo through NE release, epinephrine (E) or saline as a control was injected intraperitoneally in mature mice. E stimulates the locus coeruleus (LC) to supply NE throughout the nervous system, which enhances arousal and reduces the exploratory locomotor activity in rodents (*Carter et al., 2010*; *Liang et al., 1986*). When mice were placed in a novel environment 10 min after E-injection, wild-type, GluA3-deficient and GluA1-deficient mice showed decreased locomotion (*Figure 8A*), suggesting that LC-activity is intact in GluA3- and GluA1-deficient mice. In brain slices prepared 10 min after E-injection we observed a significant increase in mEPSC frequency in CA1 neurons of wild-type and GluA1-deficient mice compared with saline-injected littermates (*Figure 8B*). No increase in mEPSC frequency was detected when GluA1-deficient mice were injected with β-AR antagonist propranolol 20 min prior to E-injection (*Figure 8B*), indicating that potentiation of GluA2/3-currents depended on β-AR activation. In slices isolated from E-injected GluA3-deficient mice we failed to detect an increase in mEPSC events (*Figure 8B*). To assess whether synaptic potentiation upon E-injection required GluA3 in CA1 neurons, we repeated this experiment in mice that lacked GluA3 expression selectively in a subset of CA1 neurons. Three-week-old mice, whose GluA3 gene was flanked by loxP sites, (flGluA3) were stereotactically injected with AAV virus expressing either GFP-tagged Cre-recombinase (Cre-GFP) or GFP under control of the Synapsin1 promoter in the CA1 region of the hippocampus. After allowing Cre-GFP to delete the GluA3 gene and deplete GluA3 expression for three weeks, we injected E or saline intraperitoneally and measured mEPSCs on GFP-positive CA1 neurons from slices prepared 10 min after injection. Whereas the synaptic effect of E-injection was evident in slices prepared from mice injected with control GFP-virus, it was significantly reduced in neurons where GluA3 expression had been stopped out by Cre-GFP (*Figure 8C*). These data indicate that NE-release induces potentiation of GluA2/3-currents at CA1 pyramidal synapses in the hippocampus.

## Discussion

In this study we identified a novel form of synaptic plasticity in the CA1 region of the hippocampus that depends on GluA3-containing AMPARs. Historically, AMPARs of hippocampal neurons have been assumed to predominantly consist of GluA1/2s with only a small proportion of GluA2/3s. This notion was based on the observations that mRNA levels of GluA3 are 10-fold lower compared with GluA1 and GluA2 mRNA levels (*Tsuzuki et al., 2001*) and that GluA2/3s contribute little to synaptic and extrasynaptic AMPAR currents (*Andrásfalvy et al., 2003*; *Lu et al., 2009*). However, the total protein levels of GluA3 in the hippocampus were shown to be substantial (*Schwenk et al., 2014*), and it was estimated that the AMPAR population in CA1 dendrites is composed of equivalent amounts of GluA1/2s and GluA2/3s (*Kessels et al., 2009*). We here unify these seemingly contradictory findings by showing that GluA3-containing AMPARs are present at synapses and on the cell surface; however, they are electrically quiet under basal conditions.

GluA3-mediated currents substantially increase when intracellular cAMP levels are increased in CA1 neurons. Our single-channel recordings indicate that these increased currents are a consequence of an improved capacity of glutamate to open the AMPAR channel. GluA1-containing AMPARs opened their channels independently of cAMP levels, suggesting that this type of AMPAR channel plasticity is an exclusive feature of GluA2/3s. The activation of GluA2/3-plasticity by cAMP is fast, since outside-out patches were pulled from whole-cell configuration after allowing cAMP to flow inside the cell for less than ~10 s. Our experiments indicate that PKA activation is insufficient for the cAMP-driven activation of GluA2/3-plasticity but requires the activation of both PKA and Ras. A cAMP-dependent signaling pathway that depends on both PKA and Ras activation is the extracellular signal-regulated kinase (Erk) (*Li et al., 2013*; *Enserink et al., 2002*; *Ambrosini et al., 2000*). However, whether PKA either promotes Ras-Erk signaling or inhibits Erk (thereby possibly skewing Ras activation towards an alternative Ras-signaling pathway) is dependent on the levels of additional signaling factors and on cell type (*Smith et al., 2010*; *Qiu et al., 2000*). Which downstream targets of PKA/Ras activate GluA2/3-plasticity in CA1 neurons and how they alter GluA2/3-channel function remains to be established. Future structural studies may reveal whether PKA/Ras signaling triggers a putative conformational change within the GluA3 subunit that allows either glutamate to access the ligand binding site or glutamate binding to open the channel.

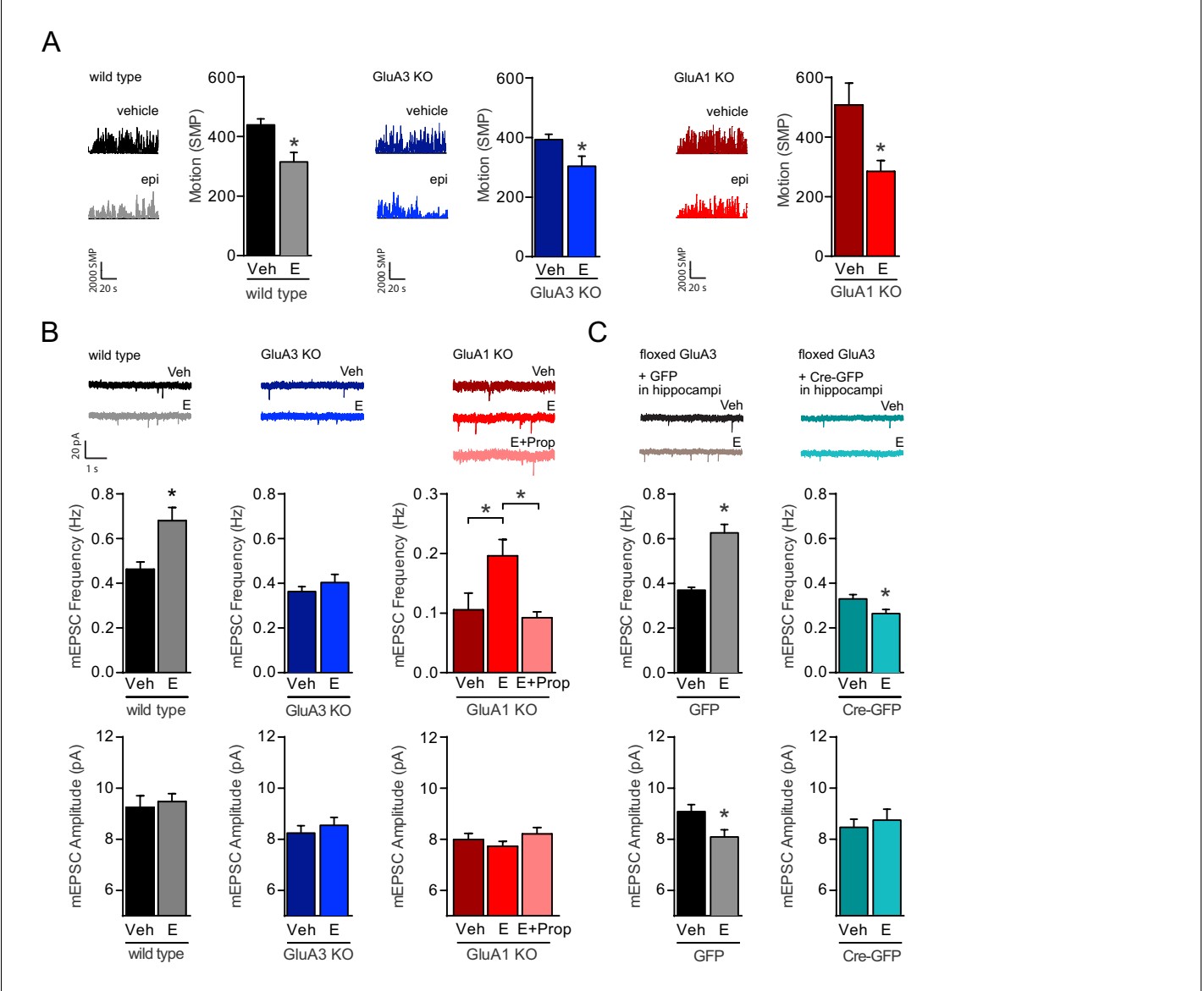

**Figure 8.** NE release triggers the activation of GluA2/3-plasticity. (**A**) Example traces and average motion as a change in significant motion pixels (SMPs) (Kopec et al., 2007) of WT injected with saline (n=15) or E (n=15), GluA3-KO injected with saline (n=17) or E (n=16), and GluA1-KO injected with saline (n=10) or E (n=11). Epinephrine (E) injection decreases the motion of mice (t-test, WT: p=0.003; GluA3-KO: p=0.02; GluA1-KO: p=0.01). (**B**) Example traces, mEPSC frequencies and mEPSC amplitudes of CA1 hippocampal neurons from WT mice injected with saline (n=13) or E (n=15) (t-test, Freq: p=0.0003; Ampl: p=0.6); GluA3-KO mice injected with saline (n=16) or E (n=13) (t-test, Freq: p=0.4; Ampl: p=0.5); or GluA1-KO mice injected with saline (n=12), E (n=20) or propranolol 20 min prior to E (n=16) (Freq: ANOVA, veh vs E: p=0.008; E vs E + Prop: p=0.04; veh vs E + Prop: p>0.9) (Ampl: ANOVA, p=0.2). (**C**) AAV virus expressing GFP or GFP-Cre were stereotactically targeted at the CA1 of flGluA3 mice. Example traces, mEPSC frequencies and mEPSC amplitudes recorded from GFP-positive CA1 neurons after injection with epinephrine (GFP: n=11; GFP-Cre: n=9) or saline (GFP: n=11; GFP-Cre: n=11). In GFP-infected neurons E-injection caused an increased mEPSC frequency (t-test, p<0.0001) and decreased amplitude (t-test, p=0.02). In GFP-Cre infected neurons E-injection caused a decrease in mEPSC frequency (t-test, p=0.02) and no change in amplitude (t-test, p=0.6). Error bars indicate SEM, * indicates p<0.05.

DOI: https://doi.org/10.7554/eLife.25462.020

The cAMP-driven activation of GluA2/3s at the surface of cell bodies mirrors GluA2/3-plasticity at synapses: both increased approximately two-fold in currents upon a rise in cAMP. These results imply that similar to its total levels at CA1 dendrites (*Kessels et al., 2009*) approximately half of the synaptic and extrasynaptic AMPAR population consists of GluA2/3s. When recombinant GluA3 was acutely expressed in GluA3-deficient CA1 neurons of organotypic slices, the newly formed GluA3-

containing AMPARs were inserted onto to the surface and into synapses without a change in synaptic or extrasynaptic AMPAR currents. However, upon a rise in cAMP they showed increased currents at both synaptic and extrasynaptic sites without a change in GluA3 levels at synapses or at the cell surface. Our results therefore indicate that GluA3-containing AMPARs constitutively traffic into synapses in an inactive state and only contribute to synaptic currents after cAMP levels have risen. In contrast, GluA1-containing AMPARs are largely kept out of the synapse under basal conditions and require LTP-like activity to traffic into synapses (*Kessels et al., 2009*; *Makino and Malinow, 2009*; *Shi et al., 2001*). A recent study showed that LTP can be expressed by the insertion of both GluA1-containing and GluA1-lacking AMPARs into synapses (*Granger et al., 2013*). Although this study did not show whether this is also the case for GluA2/3s, our finding that GluA2/3s are electrically dormant under basal conditions may explain why LTP is not visible in GluA1-deficient hippocampal slices (*Granger et al., 2013*; *Humeau et al., 2007*; *Zamanillo et al., 1999*). A previous study showed that a slowly arising LTP can be seen in GluA1-deficient slices when a theta-burst stimulation is paired with a postsynaptic depolarization (*Frey et al., 2009*). We speculate that upon this stimulation protocol a gradual increase in cAMP levels may have activated GluA2/3-plasticity.

In our experiments a postsynaptic change in AMPAR currents was reflected more prominently by a change in mEPSC frequency than by a change in mEPSC amplitude, which has been observed previously (*Watson et al., 2017*; *Lee et al., 2014*; *Lu et al., 2009*; *Rumbaugh et al., 2006*) and is a consequence of a large proportion of mEPSCs recorded from CA1 neurons falling below the 5 pA detection limit (*Figure 3G*). An increase in mEPSC frequency rather than mEPSC amplitude upon the activation of GluA2/3-plasticity may be further explained by the findings that GluA1/2s and GluA2/3s are differentially distributed at CA1 synapses. Whereas GluA3 is uniformly distributed within synapses, GluA1 tends to be more concentrated towards their edges (*Jacob and Weinberg, 2015*) and large CA1 synapses are particularly enriched in GluA1 (*Shinohara and Hirase, 2009*). When a mEPSC is generated by the release of a single vesicle onto a micro-domain within a synapse (*MacGillavry et al., 2013*) that predominantly contains GluA2/3s, its amplitude may only surpass the detection limit after a rise in cAMP. Reversely, glutamate binding to a micro-domain that contains mainly GluA1 and little GluA3 may produce mEPSCs that are detectable under basal conditions, but the amplitude of such mEPSCs will benefit only little from GluA2/3-plasticity. In such a scenario, the frequency of mEPSC events increases more than their average amplitude. Similarly, CA1 neurons, in which either GluA1 or GluA3 are depleted, are mostly affected in mEPSC frequency and little in amplitude (*Lu et al., 2009*). We therefore propose that variations in mEPSC frequency, if not caused by a change in presynaptic vesicle release or the number of functional synapses, can be a manifestation of a postsynaptic AMPAR subunit-specific effect.

Besides the hippocampus, GluA3-containing AMPARs are present in most other brain regions, including the cortex, amygdala, striatum, thalamus, brain stem, olfactory bulb, nucleus accumbens and cerebellum (*Breese et al., 1996*; *Reimers et al., 2011*; *Schwenk et al., 2014*), suggesting that GluA2/3-plasticity may be operative throughout the brain. GluA3-deficient mice show a number of physiological and behavioral abnormalities, which may putatively depend on GluA2/3-plasticity in different brain regions. For instance, GluA3-deficient mice show increased social and aggressive behavior (*Adamczyk et al., 2012*), a reduced alcohol-seeking behavior (*Sanchis-Segura et al., 2006*), impaired auditory processing (*García-Hernández et al., 2017*), and altered electroencephalographic patterns in the cortex during sleep (*Steenland et al., 2008*). Based on the findings in this study, we examined the role of GluA3-dependent plasticity at synapses onto Purkinje cells (PCs) of the cerebellum (*Gutierrez-Castellanos et al., 2017*). We found that the AMPAR-subunit specific rules for synaptic plasticity in PCs were reversed compared with CA1 neurons: motor learning and the expression of LTP did not require GluA1, but critically depended on GluA3. Similarly as at CA1 synapses, synaptic potentiation was accomplished by the activation of GluA3-channel function upon a rise in cAMP. An interesting difference is that GluA3-dependent plasticity in PCs is mediated by Epac/Rap1 (*Gutierrez-Castellanos et al., 2017*). Since Ras and Rap1 trigger similar downstream (Erk) signaling pathways with different temporal patterns (*Li et al., 2016*), Ras and Rap1 may activate a transient or persistent form of GluA2/3-plasticity respectively. Thus, GluA2/3-plasticity may have different functions contingent on the cell type and brain regions in which it is expressed.

The activation of β-ARs at CA1 neurons of the hippocampus can stimulate two independent forms of AMPAR plasticity in parallel. First, PKA phosphorylation of GluA1-containing AMPARs facilitates their trafficking to synapses (*Man et al., 2007*), thereby facilitating memory formation (*Hu et al.,*

*2007*). We identified a second cAMP-dependent form of AMPAR plasticity. Our data show that GluA2/3s at synapses increase their currents upon β-AR activation. Other signaling pathways that influence intracellular cAMP levels, like for instance those activated by dopamine, serotonin or acetylcholine release, may theoretically influence GluA3-containing AMPARs as well. It will be interesting to assess under which conditions GluA3-containing AMPARs in the hippocampus are activated and how GluA2/3-plasticity influences the formation and retrieval of contextual memories.

## Materials and methods

### Mice

The *Gria3*-deficient (GluA3-KO) and wild-type littermate colony was established from C57Bl/6 × 129P2-Gria3tm1Dgen/Mmnc mutant ancestors (RRID:MMRRC_030969-UNC) (MMRRC, Davis, CA), which were at least 6 times backcrossed to C57Bl/6 mice. *Gria1*-deficient (GluA1-KO) mice were a kind gift from Dr. R. Huganir (*Kim et al., 2005*), and a colony was generated by mating heterozygous C57Bl/6/129 mice. *Gria1xGria3* double deficient colony was established by crossing homozygote *Gria1*-deficient males with heterozygote *Gria3*-deficient females. Mice with floxed loci at the *Gria3* gene [*Gria3^{lox/lox}* (RRID:IMSR_EM:09215)] were a kind gift from Dr. R. Sprengel (*Sanchis-Segura et al., 2006*) and maintained in a homozygous colony. Mice were kept on a 12 hr day-night cycle (light onset 7am) and had *ad libitum* access to food and water. All experiments were conducted in line with the European guidelines for the care and use of laboratory animals (Council Directive 86/6009/EEC). The experimental protocol was approved by the Animal Experiment Committee of the Royal Netherlands Academy of Arts and Sciences (KNAW).

### Electrophysiology

Organotypic hippocampal slices were prepared from P7-8 mice as described previously (*Stoppini et al., 1991*) and used at 7–12 days in vitro. Where indicated, slices were infected with Sindbis virus expressing GFP- or SEP-tagged rat GluA3 (flip) 20–28 hr prior to experiments. During recordings, slices were perfused with artificial cerebrospinal fluid (ACSF; in mM): 118 NaCl, 2.5 KCl, 26 NaHCO$_3$, 1 NaH$_2$PO$_4$, supplemented with 4 MgCl$_2$, 4 CaCl$_2$, 20 glucose. Patch recording pipettes were filled with internal solution containing (in mM): 115 CsMeSO$_3$, 20 CsCl, 10 HEPES, 2.5 MgCl$_2$, 4 Na$_2$ATP, 0.4 Na-GTP, 10 Na-Phosphocreatine, 0.6 EGTA. Outside-out recordings were made with 3–5 MΩ pipettes and the bath contained the desensitization blockers PEPA (80 µM; Tocris) and cyclothiazide (100 µM; Tocris) to exclude variations due to differences in desensitization properties. Every 20 s a 100 ms puff of 100 µM S-AMPA was delivered with a Picospritzer III (Parker, Hollis, USA). Single channel recordings were measured under cell-attached configuration with 6–8 MΩ pipettes filled with internal solution to which S-AMPA (100 µM; Tocris) was added. Whole-cell recordings in organotypic slice cultures were made with 3–5 MΩ pipettes (R$_{access}$ < 20 MΩ, and R$_{input}$ > 10× R$_{access}$). During mEPSC recordings, TTX (1 µM; Tocris) and picrotoxin (100 µM; Sigma) were added to the bath. Where indicated, the following drugs were added to the perfusion solution: forskolin (50 µM; Sigma), IBMX (50 µM; Tocris), KT5720 (4 µM; Tocris), PKI (2 µM; Calbiochem), ESI05 (10 µM; Biolog); Salirasib (10 µM; Tocris); or inside the recording pipette: cAMP (100 µM; Sigma), N002 (100 µM; Biolog), 8-CPT (20 µM; Tocris). During evoked recordings, a cut was made between CA1 and CA3, and picrotoxin (50 µM) and 2-chloroadenosine (4 µM; Tocris) were added to the bath. Two stimulating electrodes, (two-contact Pt/Ir cluster electrode, Frederick Haer), were placed 100 µm apart between 100 and 300 µm down the apical dendrite and 200 µm apart laterally. AMPAR-mediated EPSCs were measured as the peak inward current at −60 mV directly after stimulation. Paired pulse ratios were determined with an inter pulse interval of 50 ms. NMDAR-mediated EPSC were measured as the mean outward current between 40 and 90 ms after the stimulation at +40 mV, and corrected by the current at 0 mV. Rectification was calculated as the ratio of the peak AMPAR current at −60 and +40 mV, corrected by the current at 0 mV, in the presence of D-APV (100 µM; Tocris) in the bath and Spermine (0.1 mM; Sigma) in the intracellular solution. EPSC amplitudes were obtained from an average of at least 30 sweeps at each holding potential. Acute hippocampal slices were prepared from 3 to 5 week-old mice. Dissection was done in ice-cold sucrose cutting solution containing (in mM): 2.5 KCl, 1.25 NaH$_2$PO$_4$, 26 NaHCO$_3$, 10 D-glucose, 230 Sucrose, 0.5 CaCl$_2$, 10 MgSO$_4$, bubbled with 95%O$_2$/5%CO$_2$. Brain slices (400 µm) were cut using a

vibratome (Thermo Scientific) and placed in a holding chamber containing ACSF supplemented with (in mM) 1 $MgCl_2$, 2 $CaCl_2$, 20 glucose and bubbled with 95%$O_2$/5%$CO_2$. They were allowed to recover at 34°C for 40 min then at room temperature for at least 40 min. Whole-cell recordings (3–5 MΩ pipettes, $R_{access}$ < 26 MΩ, and $R_{input}$ > 10 x $R_{access}$) were made in ACSF containing TTX (1 μM) and picrotoxin (50 μM) at 28°C. To block Ras, 3 μg/ml the OP01 Anti-v-H-Ras (Ab-1) Rat mAb (Y13-259, Millipore; RRID:AB_565094) or Rat IgG1 isotype control (MA1-90035, Invitrogen; RRID:AB_10984952) was included in the intracellular solution. After obtaining whole-cell configuration the antibody was allowed to diffuse in the cell for 5 min before recording. After 10 min FSK was added to the perfusion and allowed to wash in for 5 min. Data was acquired using a Multiclamp 700B amplifier (Molecular Devices). mEPSC data are based on at least 100 events or 5 min of recording, with exception of *Figure 6—figure supplement 1* (1 min). Data were analyzed with MiniAnalysis (Synaptosoft). Individual events above a 5 pA threshold were manually selected. Evoked recordings were analyzed using pClamp 10 software (Molecular Devices).

Non-stationary noise analysis of outside-out patches traces was carried out following previously described methods (*Alvarez et al., 2002*; *Hartveit and Veruki, 2007*). Peak aligned AMPA-evoked currents recorded over 10–15 sweeps per outside out patch, were binned in 10 equally sized bins of 150 ms each and for each bin, the mean amplitude and variance was calculated. The data distribution resulting after plotting amplitude versus variance was fitted with the following equation: $\sigma^2 = iI - \frac{I^2}{N} + \sigma_b^2$, where the variance ($\sigma^2$) of the amplitude of the current (I) obtained at each time point is explained as a function of the single unitary current (i) and the number of functionally conducting channels (N) with an offset given by the variance of the baseline noise ($\sigma_b^2$). From the derivative at I=0, the relative number of functional channels was extracted as well as the single channel conductance which was calculated by dividing the unitary current by the applied voltage with respect to the reversal potential ($V_{holding}$-$E_{reversal}$, −60 mV and 0 mV respectively). The peak open probability ($P_o$), corresponding to the fraction of available functional channels open at the time of the peak current ($I_{peak}$), can be calculated from the following equation: $P_0 = I_{peak}/N_{max}$, where $N_{max}$ represents the theoretical maximum of available channels opened at the point where the theoretical maximum amplitude reaches the minimum variability ($\sigma_b^2$) in the given parabola fit. Single channel activity was analyzed using ClampFit (Molecular Devices). Three detection thresholds were used to detect O1 (1.5 pA), O2 (3 pA) and O3 (4.5 pA) openings in single channel AMPARs in steady baseline recordings (no holding current fluctuations). Events with latency shorter than 0.3 ms were ignored to prevent noise from being recognized as openings. Non-stationary noise analyses for the mEPSC events were based on peak scaled mEPSCs.

## Two-photon laser scanning microscopy

Wild-type organotypic hippocampal slices were sparsely infected with Sindbis virus expressing SEP-GluA1 or SEP-GluA3 together with tdTomato as cytoplasmic marker, and were used in experiments 20–28 hr after viral infection. Three-dimensional images were collected on a custom-built two-photon microscope based on a Fluoview laser-scanning microscope (Femtonics). The light source was a mode-locked Ti:sapphire laser (Chameleon, Coherent) tuned at 910 nm using a 20× objective. During imaging, slices were kept under constant perfusion of aCSF (in mM: 118 NaCl, 2.5 KCl, 26 $NaHCO_3$, 1 $NaH_2PO_4$, supplemented with 4 $MgCl_2$, 4 $CaCl_2$, 20 glucose) at 30°C, gassed with 95% $O_2$/5%$CO_2$. Images were captured every 1 μm from infected CA1 pyramidal cell bodies or apical dendrites past the point of bifurcation of primary to secondary dendrites, approximately 300 μm from the cell body. Fluorescence intensity was quantified from projections of stacked sections using ImageJ software (NIH). For photobleaching experiments, apical dendrites were imaged 100–300 μm from the cell body (pixel size x,y,z 0.1 × 0.1 × 0.5 μm). Photobleaching of SEP-fluorescence on spines was achieved by prolonged xy scanning for 30 s. The cytoplasmic tdTomato signal recovered immediately after photobleaching (*Figure 4—figure supplement 1*). To determine the photon recovery after photobleaching (FRAP), background-subtracted and leak-corrected red and green fluorescence were quantified and the mean signal intensity of spines was normalized to that of the dendrite and compared across time.

## Stereotactic hippocampal viral injections and E-injection

Adeno-associated virus (AAV) with a titer between $10^{12}$–$10^{13}$ particles/ml were produced from AAV5-pSynapsin1-GFP and AAV5-pSynapsin1-CreGFP. 3 week-old *Gria3*<sup>lox/lox</sup> mice were anesthetized with isofluorane (induction 5%, maintenance 2%) and positioned in a stereotaxic apparatus, kept on a heating pad. Bilateral hippocampal injections of viral solutions (3 injection sites per side; 400 nl per site; AP: −1.5, −1.7, −1.9; L: ±1.5; DV: 1.2 mm) were delivered with a glass micropipette through a hole drilled in the skull by pressure application (Nanoject II, Auto-Nanoliter Injector, Drummond Scientific). E-injection experiments were performed with *Gria3*<sup>lox/lox</sup> mice 3 weeks after viral injections, or with WT, GluA3-KO or GluA1-KO littermate mice at 3–4 weeks of age. (±)-Epinephrine hydrochloride (0.5 mg/kg, Sigma-Aldrich) and (±)-Propranolol hydrochloride (20 mg/kg, Sigma-Aldrich) were dissolved in saline (0.9%, NaCl) and injected intraperitoneally (5 ml/kg). 10 min after E-injection mice were place in a novel environment for 2 min and the locomotion of the mice was measured as previously described (*Kopec et al., 2007*) or they were sacrificed to acutely isolate brain slices.

## Statistics

A power analysis was performed prior the experiments to estimate the average sample size. For power analyses 2-sample, two-sided tests were used with the assumption of equal variance, a power of 0.8 and a Type I error (alpha) of 0.05 were used. If effect sizes could not be estimated based on prior experiments a minimal effect size of 0.2 was used. For all experiments biological replication of effect was obtained by the use of slices from at least three different litters of mice. Experiments that are depicted in the same graph were performed in parallel and with hippocampal slices of littermate mice. Data sets were Log-transformed and normal distributions were obtained. All data were analyzed using two-tailed Student *t* tests to compare 2 conditions (unpaired, unless indicated paired t-test) or with ANOVAs with post-hoc Tukey comparisons for more than 2 conditions. Reported p-values are post hoc contrasts, unless overall ANOVA was not significant. Repeated-Measures ANOVAs were used to assess effects over time. *P* values below 0.05 were considered statistically significant. For source data, please see Renner et al Source_data.xlsx.

## Acknowledgements

We thank Wobbie van den Hurk and Dr. Karlijn van Aerde for technical assistance, Dr. Hans Bos for offering PKA and Epac specific drugs, and Dr. Chris de Zeeuw, Dr. Hailan Hu, Dr. Christian Lohmann, and Dr. Christiaan Levelt for helpful comments on the manuscript. This work was supported by the Netherlands Organization for Scientific Research (HWK).

## Additional information

### Funding

| Funder | Grant reference number | Author |
| --- | --- | --- |
| Nederlandse Organisatie voor Wetenschappelijk Onderzoek | 821.02.016 | Helmut W Kessels |
| Nederlandse Organisatie voor Wetenschappelijk Onderzoek | 864.11.014 | Helmut W Kessels |

The funders had no role in study design, data collection and interpretation, or the decision to submit the work for publication.

### Author contributions

Maria C Renner, Conceptualization, Data curation, Formal analysis, Investigation, Writing—original draft; Eva HH Albers, Conceptualization, Data curation, Formal analysis, Investigation, Writing—original draft, Writing—review and editing; Nicolas Gutierrez-Castellanos, Data curation, Formal analysis, Investigation, Methodology; Niels R Reinders, Data curation, Formal analysis, Investigation; Aile N van Huijstee, Data curation, Formal analysis; Hui Xiong, Tessa R Lodder, Data curation; Helmut W

Kessels, Conceptualization, Data curation, Formal analysis, Supervision, Funding acquisition, Validation, Investigation, Writing—original draft, Writing—review and editing

**Author ORCIDs**
Eva HH Albers ⬤ http://orcid.org/0000-0002-3956-9494
Helmut W Kessels ⬤ http://orcid.org/0000-0002-1122-745X

**Ethics**
Animal experimentation: All experiments were conducted in line with the European guidelines for the care and use of laboratory animals (Council Directive 86/6009/EEC). The experimental protocol was approved by the Animal Experiment Committee of the Royal Netherlands Academy of Arts and Sciences (KNAW).

**Decision letter and Author response**
Decision letter https://doi.org/10.7554/eLife.25462.022
Author response https://doi.org/10.7554/eLife.25462.023

## Additional files

**Supplementary files**
• Source data 1. File containing datapoints of all figures.
DOI: https://doi.org/10.7554/eLife.25462.021

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
