## [Decision Letter]

[Editors’ note: this article was originally rejected after discussions between the reviewers, but the authors were invited to resubmit after an appeal against the decision.]

Thank you for submitting your work entitled "Synaptic Plasticity through activation of AMPA-receptor subunit GluA3" for consideration by *eLife*. Your article has been reviewed by three peer reviewers, and the evaluation has been overseen by a Reviewing Editor and a Senior Editor. The reviewers have opted to remain anonymous. Our decision has been reached after consultation between the reviewers. Based on these discussions and the individual reviews below, we regret to inform you that your work will not be considered further for publication in *eLife*.

Summary:

The reviewers and editors all recognized the interesting core result regarding the increase in open probability of hippocampal AMPA receptors upon exposure to PKA. They also noted, however, a number of technical questions, internal inconsistencies, relatively small sample sizes with respect to the variance, and possible alternative interpretations. Because it would take substantial additional experiments to address these points and resolve all the associated questions, we cannot consider this work further. (Although some points in the reviews brought up new types of experiments and the relation of this study to previous work, these issues were not the basis for our decision.) The full reviews are included below.

Reviewer #1:

The manuscript by Renner et al. tests whether increasing cAMP levels in pyramidal neurons reveals a role of GluA3 in synaptic transmission. Understanding the mechanisms by which second messengers like cAMP regulate the properties of AMPA receptors and fast synaptic transmission is critical. The authors by using a series of electrophysiological and imaging experiments argue that under basal conditions GluA3 channels have very little contribution in synaptic transmission due to their low conductance and probability opening. However, when cAMP levels are elevated GluA3 shifts to a high-conductance state allowing them to contribute to point-to-point transmission in hippocampus. Importantly, this shift in the GluA3 mode is not due to PKA or EPAC, two well-established effectors of cAMP, but rather through Ras activation.

Although the authors' findings provide a new insight in regards to the regulation of AMPA receptors in hippocampus, there are concerns with this work.

1) Throughout, the authors use FSK or direct application of high levels of cAMP to test whether cAMP can increase the activity of AMPA receptors and in this work GluA3. It would have been important to also test whether the same effect occurs when cAMP is elevated through β-adrenergic receptors, a more physiologically relevant means to increase cAMP levels in hippocampus. In general it's unclear whether the effects are a result of a very large cAMP stimulus.

2) They then argue that cAMP exerts its effects through Ras. Considering that, they also show that cAMP can induce its effects in outside-out patches, – is Ras associated with GluA3? How do the authors know whether Ras directly activates GluA3 channels rather than regulate the relationship between TARPs or cornichons and GluA3; auxiliary AMPA receptor subunits known to control the biophysical properties of AMPA receptors? These questions should have been addressed in order to provide a mechanistic insight on how cAMP-Ras control AMPA receptors. Importantly, it would have been important to test the proposed cAMP and Ras mechanism in an expression system that does not natively express AMPA receptors, like HEK293 cells.

3) The authors argue that application of FSK in slices increased the AMPA and NMDA ratio presumably due to potentiation of postsynaptic GluA3 receptors, as GluA3 null neurons do not show this increase in the AMPA/NMDA ratio. They argue against any presynaptic effects of FSK, as PPR did not change in the presence of FSK (Figure 3). However, they then show that FSK increases mEPSC frequency in the GluA3 null neurons, although not to the same extent as in control neurons. This would then suggest that FSK has presynaptic effects, which are not controlled in this work. Additionally, previous work has also shown that cAMP changes NMDA receptor activity (Raman et al. 1996; Westphal et al. 1999), as such the AMPA/NMDA ratio is difficult to quantitatively interpret in the presence and absence of FSK/cAMP.

Reviewer #2:

In this manuscript Renner et al. reported that cAMP/FSK enhances GluA3-containing AMPAR-mediated currents in hippocampal neurons, similar to their finding in cerebellar Purkinje cells published earlier this year in Neuron (Gutierrez-Castellanos et al., 2017). Using outside-out patches and single channel recording of somatic membrane of CA1 neurons derived from WT, GluA1 or GluA3 KOs, they concluded that this enhancement is due to an increase in open probability of GluA3-containing AMPARs. Similar enhancements were observed at the synaptic level when analyzing AMPAR-mediated mEPSCs. The authors further explored signaling pathways underlying cAMP/FSK enhanced synaptic currents pharmacologically using inhibitors. This manuscript identifies a novel regulation of GluA3-containing AMPARs. However, due to the complexity of the effects of cAMP and some internal discrepancies in the data it is difficult to come to the simple conclusions presented by the authors. Specific comments are listed as the following:

1) Assuming AMPARs in hippocampus are predominately in forms of GluA1/2 and GluA2/3, the authors compared cAMP/FSK-induced currents in WT (which contains both GluA1/2 and GluA2/3), GluA1 KO (which contains GluA2/3 only) and GluA3 KO (which contains GluA1/2 only). Since the GluA3 KO cells did not respond to cAMP/FSK like the WT and GluA1 KO did, the authors concluded that cAMP/FSK selectively potentiate currents from GluA"2/3" type of AMPARs. The Effect of cAMP could actually be on GluA2 when combined with GluA3. The title of this manuscript "Synaptic Plasticity through activation of AMPA-receptor subunit GluA3" is therefore somewhat misleading, as they did not show a direct effect of cAMP/FSK on GluA3 subunit nor provide its role in synaptic plasticity. Further evidence is required to make GluA3-specific claims and its effects on synaptic plasticity.

2) The trafficking experiments done by live imaging of SEP-GluA3 delivered by sindbis virus (Figure 1), as well as transfected SEP-GluA1 or SEP-GluA3 in CA1 neurons (Figure 4) are not convincing as these subunits when overexpressed alone tend to form homomers (GluA1/1 or GluA3/3) and may not reflect trafficking events of endogenous AMPARs (Sin et al., 2001). GluA3/3 in particular were reported to be absent in synapses even though it can be observed in spines. Alternative methodology examining endogenous AMPAR subunits such as isolation of PSD fraction of AMPARs or immunostaining of synaptic AMPARs would be a better approach.

3) AMPA/NMDA ratio results are confusing. In Figure 3—figure supplement 1, the major effect of FSK in WT cells appears to decrease NMDAR current, but leave AMPAR current unchanged. If so, what is the explanation? Is loss of the FSK effect on A/N ratio in GluA3 KO NMDAR-related?

4) The rescue experiment, in which GFP-GluA3 is reintroduced into GluA3 KO to evaluate GluA3-specific manipulation in synaptic transmission, is not convincing. While the FSK-induced frequency increase in WT cells is ~4 fold (1 to 4 Hz), the magnitude in GluA3 KO is ~2 fold (1 to 2Hz, Figure 1) similar to the magnitude obtained from the rescue condition (~ 2 fold, 0.75 to 1.5 Hz, Figure 1, GFP-GluA3 infected). The author should plot Figure 3 using the same scale for the ease of comparison of the frequency graphs. Similarly, the absence of FSK-induced potentiation in amplitude in GluA3 KO cells (Figure 1, GluA3 KO) was not rescued in GFP-GluA3 infected GluA3 KO neurons (Figure 1, GFP-GluA3 infected).

5) FSK is well known to potentiate GluA1-containing AMPARs and FSK induced mEPSC frequency can be observed in WT, GluA3 KO and GluA1 KO neurons (Figure 3), the pharmacology and antibody effects on mEPSC frequency done in WT cells (all experiments in Figure 6) are difficult to interpret. Further experiments using KO neurons are required to make a clear claim.

6) In Figure 6, the basal (FSK-/PKI+) mEPSC frequency in RAS-IgG neurons seems to be higher than the Ctrl-IgG. However, the basal (FSK-/PKI-) mEPSC frequency seems comparable between Ctrl-IgG and RAS-IgG neurons. If RAS-IgG neurons with PKI treatment do show a higher frequency than the Ctrl-IgG neurons, this would be an occlusion effect of PKI but not a blocking effect of RAS-IgG.

Reviewer #3:

Renner et al. propose a novel mechanism of cAMP-mediated AMPA receptor current potentiation. By using AMPA receptor knockout mice, they dissect the contribution of the two AMPA subunits GluA1 and GluA3 to this effect. With recordings of extrasynaptic currents they show that cAMP alters conductance states of Glu3 but not of GluA1 channels. Recordings of synaptic currents and FRAP experiments suggest that this increase in GluA3 conductance rather than increased insertion of AMPA receptors into synapses is responsible for the cAMP-mediated synaptic potentiation. Finally, they describe that RAS activity is required for the cAMP-mediated synaptic potentiation.

The study is interesting as it identifies how "dormant" GluA3 channels are activated by cAMP. This finding, however, is not novel. The authors published this year another study on a very similar mechanism in Purkinje cells of the cerebellum (Gutierrez-Castellanos et al., Neuron 2017). There are some differences in the findings of the two studies (e.g. Epac vs. Ras dependency), but the main novelty of the current study is that cAMP influences GluA3 also in the hippocampus. Similar to the previous study, the authors provide high quality results of several experiments that very convincingly show the influence of cAMP on extrasynaptic GluA3 channels. However, results of extrasynaptic recordings are less convincing (see below). Finally, it would be important to show if and how much a cAMP-mediated increase in GluA3 currents contributes to a more physiological form of synaptic plasticity. In conclusion, the study is still preliminary in its present form and needs additional experiments for publication in *eLife*.

1) Variability of mEPSC frequencies across experiments is very high. This precludes comparing results from the different experiments and drawing conclusions about the contribution of GluA1 or GluA3 from those comparisons. For example, mEPSC frequency in wildtype neurons without treatment differ by 0.7 Hz (Figure 5: 0.7; Figure 3: 1.4 Hz). In addition, the FSK-mediated increase is very different in wildtype neurons (+3.2 Hz in Figure 3 and 1.2 Hz in Figure 5). In fact, the FSK-mediated increase shown in Figure 5 is similar to the increase in GluA3 knockout mice shown in Figure 3 (+ 1 Hz). Results shown in Figure 3 and Figure 5 strongly suggest that GluA3 channels play a role for the increase in synaptic strength in wildtype neurons. However, the contribution is most likely small (Figure 3 +0.6 Hz, Figure 5 +0.3 Hz). Again, the high variability of mEPSC frequency in the different experiments makes a direct comparison very difficult. Thus, my major concern is that experiments shown in Figure 3, Figure 5 and Figure 6 do not allow drawing conclusions about the relevance of cAMP-mediated increase in GluA3 function for synaptic plasticity in CA1 neurons. That is for example in contrast to the study of Gutierrez-Castellanos et al. (Neuron 2017) that convincingly showed that GluA3 channels and the novel GluA3 plasticity play a role for synaptic function of Purkinje cells (mEPSC frequency, amplitude, LTP) and mouse behaviour (VOR).

2) What was the rational to use acute brain slices from 3-5 weeks old mice in some experiments and organotypic slices of newborn mice in others? The mechanisms of FSK potentiation of synaptic currents might very well differ in the two preparations.

3) The conclusion that Ras plays a role in GluA3-plasticity and competes with PKA-mediated GluA1-plasticity is not convincing. Except for the small increase in mEPSC amplitude, Ras-IgG have no influence on mEPSCs. In addition, experiments in Figure 6 are again very difficult to interpret since variability is high. Thus, is the lack of an effect of FSK explained because RAS-IgG increases mEPSC frequency in the -FSK control condition? mEPSC frequency in the presence of FSK is very similar irrespective of presence or absence of PKI or RAS-IgG (Figure 6). In addition, what was the rational to compare Fold-changes in frequency in Figure 6 instead of mEPCS frequency as in all other experiments? Finally, the number of neurons analyzed should be increased. In fact, I am surprised that the small difference in Fold-change in 6C reaches significance with an N of 5 and I am pretty sure that there would be no significant difference in the FSK-mediated increase in mEPSC frequency between IgG and RAS-IgG.

[Editors’ note: what now follows is the decision letter after the authors submitted for further consideration.]

Thank you for resubmitting your work entitled "Synaptic Plasticity through activation of GluA3-containing AMPA-receptors" for further consideration at *eLife*. Your revised article has been favorably evaluated by Gary Westbrook (Senior Editor), a Reviewing Editor, and two reviewers.

The manuscript has been improved but there are some remaining issues that need to be addressed before acceptance, as outlined below:

Remaining revisions:

Both reviewers acknowledged the value of the additional experiments but both brought up areas in which the interpretation of experiments should be constrained or clarified. Specifically,

1) The experiments directly activating β-adrenergic receptors are interesting but the experiments were done under extreme (with IBMX) or nonspecific (with i.p.) conditions. The reviewers recognize that these are useful as proof of principle, but we suggest that the limitations of the experiment be made clear, particularly in discussions of the literature (e.g. extent to which these results can reconcile other studies).

2) Also, the analysis of mEPSCs raised questions regarding the specificity/magnitude of the GluA3 contribution to the effect of blocking RAS/PKA. Please interpret/explain the data in context of the difference in baseline rates between wild-type and KO, and curtail and/or clarify the conclusions accordingly.

The full comments of the reviewers are given below to place the comments in context. We expect that these points can be addressed with textual/figure revisions.

Reviewer #1:

This is a revised manuscript by Renner et al. showing that high concentrations of cAMP facilitates AMPA receptors through GluA3 subunits. The authors have performed several new experiments that clarify and support their previous conclusions. However, one of my previous concerns was the use of very high concentrations of cAMP or the use of FSK. To address this they tested whether activation of β-adrenergic receptors – a more physiologically relevant experiment – can also lead to GluA3 mediated plasticity. They indeed demonstrate that application of ISO evokes GluA3 potentiation, however only when PDEs are blocked. This suggests that the proposed GluA3 plasticity might take place under non-physiological conditions. They also show that i.p. injection of EPI in mice could lead to GluA3 AMPA receptor potentiation. This experiment is very difficult to interpret, as global activation of b-adrenergic receptors will initiate multiple downstream signaling pathways. Due to these reasons my enthusiasm for this work is still dampened.

Reviewer #3:

Renner and colleagues added a substantial number of new experiments that clarified some concerns. In particular the experiments shown in Figure 7 increased the novelty of the study and provided evidence that physiological stimuli can induce cAMP-mediated activation of GluA3-containing AMPA-receptors. In conclusion, the comprehensive study of Renner et al. is largely convincing and suitable for publication in *eLife*.

I am still not convinced by the mEPSC data as the high variability of mEPSC frequency within one genotype precludes drawing conclusions about the contribution of GluA3 during cAMP mediated changes in synaptic function. A calculation of fold-change in frequency is not really helpful for an estimation of the contribution of GluA3. Absolute changes in frequency have to be compared for such an estimation. The very small increase in mEPSC frequency in GluA1 KO mice (approx. 0.1 Hz in Figure 6) compared to the much bigger increase in wt mice (approx. 0.5 Hz also in Figure 6) suggests that the contribution of GluA2/3 heteromers is rather small. Please discuss.

---

## [Author Response]

[Editors’ note: the author responses to the first round of peer review follow.]

Reviewer #1:

[…] Although the authors' findings provide a new insight in regards to the regulation of AMPA receptors in hippocampus, there are concerns with this work.

1) Throughout, the authors use FSK or direct application of high levels of cAMP to test whether cAMP can increase the activity of AMPA receptors and in this work GluA3. It would have been important to also test whether the same effect occurs when cAMP is elevated through β-adrenergic receptors, a more physiologically relevant means to increase cAMP levels in hippocampus. In general it's unclear whether the effects are a result of a very large cAMP stimulus.

In the revised manuscript we have added a new experiment that indicates that the activation of β-adrenergic receptors by isoproterenol triggers GluA3-dependent synaptic potentiation in CA1 neurons in brain slices (Figure 7). In addition, we show that NE-release in the hippocampus induced by i.p. epinephrine injection leads to GluA3-dependent synaptic potentiation in CA1 neurons (Figure 8).

2) They then argue that cAMP exerts its effects through Ras. Considering that, they also show that cAMP can induce its effects in outside-out patches, – is Ras associated with GluA3? How do the authors know whether Ras directly activates GluA3 channels rather than regulate the relationship between TARPs or cornichons and GluA3; auxiliary AMPA receptor subunits known to control the biophysical properties of AMPA receptors? These questions should have been addressed in order to provide a mechanistic insight on how cAMP-Ras control AMPA receptors. Importantly, it would have been important to test the proposed cAMP and Ras mechanism in an expression system that does not natively express AMPA receptors, like HEK293 cells.

When obtaining outside-out patches, cAMP is present in the recording pipette and allowed to diffuse into the cell body upon achieving whole-cell configuration for up to ~10 seconds. Within this time frame, outside-out patches were pulled, which showed increased AMPAR currents. Thus, within seconds cAMP triggered intracellular signaling leading to increased AMPAR currents. When we repeated this experiment without adding cAMP in the recording pipette, but instead we pulled inside-out patches and used direct application of cAMP onto these inside-out patches, we did not observe an activation of GluA3-dependent plasticity (Figure 5—figure supplement 2). This experiment argues against cAMP directly activating proteins associated with the AMPAR complex.

We have expanded our experiments on the dependency of Ras of the activation of GluA3-plasticity (Figure 6 in revised manuscript). These experiments reveal that both Ras and PKA activation need to be blocked to prevent synaptic potentiation of GluA3-containing AMPARs. We discuss the implications of these findings in the second paragraph of the Discussion section.

The PKA/Ras signaling pathway has been described previously in non-neuronal cells. Our study is the first to show a functional consequence of PKA/Ras signaling in neurons. How PKA/Ras signaling leads to the activation of GluA2/3 channels we will address during the next years in collaboration with experts on PKA/Ras signaling and with experts on AMPAR structure/function relationships.

3) The authors argue that application of FSK in slices increased the AMPA and NMDA ratio presumably due to potentiation of postsynaptic GluA3 receptors, as GluA3 null neurons do not show this increase in the AMPA/NMDA ratio. They argue against any presynaptic effects of FSK, as PPR did not change in the presence of FSK (Figure 3). However, they then show that FSK increases mEPSC frequency in the GluA3 null neurons, although not to the same extent as in control neurons. This would then suggest that FSK has presynaptic effects, which are not controlled in this work. Additionally, previous work has also shown that cAMP changes NMDA receptor activity (Raman et al. 1996; Westphal et al. 1999), as such the AMPA/NMDA ratio is difficult to quantitatively interpret in the presence and absence of FSK/cAMP.

The main conclusion of our manuscript is that a rise in cAMP increases postsynaptic GluA3-dependent currents. That a rise in cAMP besides postsynaptic changes, also produces presynaptic plasticity, is not challenged by us. Nevertheless, in our experimental preparations forskolin predominantly produced postsynaptic effects in CA1 neurons, which is in line with a previous study (Sokolova et al., 2006). This conclusion is based on the following observations/analyses:

- Forskolin produced an increase in AMPA/NMDA ratio in WT neurons, but not in GluA3-KO neurons. A variance analysis on our recordings of AMPA/NMDA ratios in WT neurons (Figure 5—figure supplement 2) indicates that:

a) forskolin did not detectably increase the presynaptic release probability;

b) forskolin increased AMPAR currents without a change in NMDAR currents.

This analysis indicates that the effects of forskolin on synapses in our experiments were in large part postsynaptic in nature. The detailed explanation of these conclusions based on the variance analysis is provided in the legend of Figure 5—figure supplement 2.

- A change in mEPSC frequency does not necessarily reflect a change in presynaptic plasticity, but can also reflect a postsynaptic change in AMPAR currents as shown in previous studies (e.g.: Watson et al., 2017; Lee et al., 2014; Lu et al., 2009; Rumbaugh et al., 2006). We plotted the distribution of mEPSC amplitudes recorded from GluA1-KO neurons (Figure 3), and this suggests that a substantial proportion of mEPSC events fell below the 5 pA detection limit. In the presence of forskolin the proportion that reached above detection limit appeared to become larger (i.e. a rightward shift in Figure 3). A further explanation of this notion is provided in the fourth paragraph of the Discussion section.

- We added a non-stationary noise analysis of scaled mESPCs in Figure 3 of the revised manuscript. This analysis demonstrates that forskolin triggered an increase in postsynaptic single-channel conductance at synapses only when neurons express GluA3.

Reviewer #2: […] 1) Assuming AMPARs in hippocampus are predominately in forms of GluA1/2 and GluA2/3, the authors compared cAMP/FSK-induced currents in WT (which contains both GluA1/2 and GluA2/3), GluA1 KO (which contains GluA2/3 only) and GluA3 KO (which contains GluA1/2 only). Since the GluA3 KO cells did not respond to cAMP/FSK like the WT and GluA1 KO did, the authors concluded that cAMP/FSK selectively potentiate currents from GluA"2/3" type of AMPARs. The Effect of cAMP could actually be on GluA2 when combined with GluA3. The title of this manuscript "Synaptic Plasticity through activation of AMPA-receptor subunit GluA3" is therefore somewhat misleading, as they did not show a direct effect of cAMP/FSK on GluA3 subunit nor provide its role in synaptic plasticity. Further evidence is required to make GluA3-specific claims and its effects on synaptic plasticity.

To address this issue, we expressed GFP-GluA2Q in GluA3-deficient neurons. GluA2Q forms predominantly GluA2/2 homomers and constitutively traffics into synapses, similar to GluA2/3s (Shi et al., 2001; Makino and Malinow, 2011). Notably, GluA2Q-expressing neurons reacted similarly to forskolin compared with GluA3-expressing ones: both showed an increased mEPSC frequency upon forskolin application to a similar extent (see Author response image 1).

These data suggest that the cAMP-driven increase in AMPAR-channel conductance is not an exclusive feature of GluA3, but may also be held by GluA2. This is perhaps not surprising, considering that GluA2 and GluA3 are quite homologous in amino acid composition and structure. GluA1/2 heteromers do not show an increase in channel conductance upon a rise in cAMP, implying that GluA1 may suppress the ability of GluA2 to increase currents when cAMP levels rise. However, this needs further investigation. Because we feel this further investigation falls beyond the scope of this manuscript, we decided to not include this experiment in the current manuscript. However, if the reviewers prefer, we are happy to include this figure in the revised manuscript.

Based on this experiment, we have adapted our wording throughout the manuscript. We have changed the title into: “Synaptic Plasticity through Activation of GluA3-containing AMPA-receptors”. In addition, through the revised manuscript we use the terminology of either ‘GluA3-dependent plasticity’ or ‘GluA2/3-plasticity’, instead of ‘GluA3-plasticity’.

**Author response image 1. respfig1:** mEPSC frequency and amplitude of GluA3-deficient neurons, infected with GFP-GluA3 (blue) or GFP-GluA2Q (yellow) in the absence (even) or presence (blocked bar) of forskolin.

2) The trafficking experiments done by live imaging of SEP-GluA3 delivered by sindbis virus (Figure 1), as well as transfected SEP-GluA1 or SEP-GluA3 in CA1 neurons (Figure 4) are not convincing as these subunits when overexpressed alone tend to form homomers (GluA1/1 or GluA3/3) and may not reflect trafficking events of endogenous AMPARs (Sin et al., 2001). GluA3/3 in particular were reported to be absent in synapses even though it can be observed in spines. Alternative methodology examining endogenous AMPAR subunits such as isolation of PSD fraction of AMPARs or immunostaining of synaptic AMPARs would be a better approach.

We agree that it would be preferable if we could complement our data with alternative methodology like immunohistochemistry. In collaboration with the lab of Prof. Guus Smit, we have tested a set of commercially available GluA3 antibodies in immunohistochemistry and immunoprecipitation experiments. Unfortunately, all of these antibodies showed substantial staining in GluA3-KO samples and showed cross-reactivity in immunoprecipitation experiments.

To assess whether the overexpression of GluA3 leads to the formation of GluA3/3 homomers, we expressed GFP-GluA3 in GluA3-KO neurons through sindbis infection and measured the rectification index of AMPAR currents in the presence of FSK. The AMPAR currents were in fact significantly *less* rectifying compared with uninfected GluA3-deficent neurons in the same slices. This experiment indicates that overexpressed GFP-GluA3 predominantly reached synapses in the configuration of GluA2/3 heteromers. This new experiment is added to the revised manuscript in Figure 3—figure supplement 3.

With respect to SEP-GluA1 overexpression: in an imaging experiment similar to our experiment in Figure 4, it was previously shown that GluA1/1 homomers behave similarly as GluA1/2 heteromers with respect to synaptic trafficking (Makino and Malinow, 2011).

Importantly, our results obtained with AMPAR subunit overexpression match and complement our results from AMPAR subunit knockout models.

3) AMPA/NMDA ratio results are confusing. In Figure 3—figure supplement 1, the major effect of FSK in WT cells appears to decrease NMDAR current, but leave AMPAR current unchanged. If so, what is the explanation? Is loss of the FSK effect on A/N ratio in GluA3 KO NMDAR-related?

We apologize for the poor explanation of the variance analysis in Figure 3—figure supplement 1. This experiment in fact demonstrates that FSK did not change NMDAR currents.

In this experiment we stimulated WT neurons to achieve AMPAR responses of approximately 50 pA. At this stimulation strength, NMDAR currents were subsequently measured. These were significantly smaller in the FSK condition, resulting in an increased AMPA/NMDA ratio. This can be the result of 1) increased AMPAR currents or 2) decreased NMDAR currents. To assess whether the increased AMPA/NMDA ratio is a consequence of increased AMPAR currents or decreased NMDAR currents, we performed a variance analysis to determine the quantal content (QC) of glutamate release. The QC can be calculated based on the coefficient of variation (CV) and depends on both presynaptic release probability (Pr) and on the number of active synapses (N) that are stimulated, but not on postsynaptic quantal size (q), as shown in the following equation:

1/CV^[2]^ = N x Pr/(1-Pr)

In the presence of FSK, a significantly lower QC is sufficient to achieve 50 pA AMPAR responses. However, the NMDAR responses are lower proportional to a lower QC in the presence of FSK (see Figure 3—figure supplement 1).

Interpretation of these results:

A lower QC can mean either a lower Pr, or a lower N. Irrespectively, these data indicate that forskolin increased postsynaptic AMPAR strength, since AMPAR amplitude of 50 pA were achieved despite a lower P_r_ or N.

It is unlikely that FSK substantially *lowered* presynaptic release probability, since we show that forskolin led to synaptic potentiation expressed by increased mEPSC frequency and amplitude (Figure 3).

Thus, the change in QC is most likely reflects a change in ‘N’. Specifically, fewer synapses (approximately two-fold lower N) were stimulated to reach 50 pA AMPAR responses. When stimulating these ~2-fold fewer synapses, ~2-fold lower NMDAR currents were measured.

Based on this analysis we propose that FSK led to an on average ~2-fold increase in postsynaptic AMPAR currents, without a significant change in NMDAR currents and without an increase in presynaptic release probability.

4) The rescue experiment, in which GFP-GluA3 is reintroduced into GluA3 KO to evaluate GluA3-specific manipulation in synaptic transmission, is not convincing. While the FSK-induced frequency increase in WT cells is ~4 fold (1 to 4 Hz), the magnitude in GluA3 KO is ~2 fold (1 to 2Hz, Figure 1) similar to the magnitude obtained from the rescue condition (~ 2 fold, 0.75 to 1.5 Hz, Figure 1, GFP-GluA3 infected). The author should plot Figure 3 using the same scale for the ease of comparison of the frequency graphs. Similarly, the absence of FSK-induced potentiation in amplitude in GluA3 KO cells (Figure 1, GluA3 KO) was not rescued in GFP-GluA3 infected GluA3 KO neurons (Figure 1, GFP-GluA3 infected).

We generally observe variations in average basal transmission of CA1 neurons between slices from different animals. In order to make comparisons across different experiments, we therefore normalize the experimental effects to their matched control conditions.

In the revised manuscript we only use data in which the experimental condition (e.g. cAMP application, forskolin, isoproterenol) has been performed in the same sets of slices as the control condition recorded on the same day, and in each experiment we use slices from multiple animals. In the original Figure 3, the WT and GluA3-KO inadvertently contained mEPSC recordings in which the ctrl and forskolin condition were obtained from different animals. We have corrected this in the revised manuscript. Due to the variation in basal transmission we assess the ability of cAMP to increase in synaptic transmission in different genotypes by comparing the fold change of the forskolin condition compared to control condition.

In the rescue experiment, the effect of GFP-GluA3 expression was tested as soon as 24 hrs after viral infection, and at this time point the forskolin-driven fold increase in mEPSC frequency was already partially rescued. The rescue experiment is particularly well-controlled since all conditions (GluA3-KO with and without GFP-GluA3 and with and without forskolin) were recorded on the same day on the same preparations of slices.

5) FSK is well known to potentiate GluA1-containing AMPARs and FSK induced mEPSC frequency can be observed in WT, GluA3 KO and GluA1 KO neurons (Figure 3), the pharmacology and antibody effects on mEPSC frequency done in WT cells (all experiments in Figure 6) are difficult to interpret. Further experiments using KO neurons are required to make a clear claim.

We have expanded the experiments with antiRas-IgG infusion in WT neurons and have repeated these experiments in GluA1-KO neurons, as shown in Figure 6 of the revised manuscript. These new experiments now show that the forskolin-driven potentiation of GluA2/3 currents depends on the activation of both Ras and PKA. The blockade effect of Ras and PKA was indeed most pronounced when GluA2/3 currents are isolated in GluA1-KO slices.

*6) In Figure 6, the basal (FSK-/PKI+) mEPSC frequency in RAS-IgG neurons seems to be higher than the Ctrl-IgG. However, the basal (FSK-/PKI-) mEPSC frequency seems comparable between Ctrl-IgG and RAS-IgG neurons. If RAS-IgG neurons with PKI treatment do show a higher frequency than the Ctrl-IgG neurons, this would be an occlusion effect of PKI but not a blocking effect of RAS-IgG.*

We have analyzed the effects of infusion of antiRas-IgG and ctrl-IgG and of PKI on basal transmission in both WT and GluA1-KO CA1 neurons, and this analysis shows that they do not significantly affect synaptic currents, as shown in Figure 6—figure supplement 1 of the revised manuscript.

Reviewer #3: […] 1) Variability of mEPSC frequencies across experiments is very high. This precludes comparing results from the different experiments and drawing conclusions about the contribution of GluA1 or GluA3 from those comparisons. For example, mEPSC frequency in wildtype neurons without treatment differ by 0.7 Hz (Figure 5: 0.7; Figure 3: 1.4 Hz). In addition, the FSK-mediated increase is very different in wildtype neurons (+3.2 Hz in Figure 3 and 1.2 Hz in Figure 5). In fact, the FSK-mediated increase shown in Figure 5 is similar to the increase in GluA3 knockout mice shown in Figure 3 (+ 1 Hz). Results shown in Figure 3 and Figure 5 strongly suggest that GluA3 channels play a role for the increase in synaptic strength in wildtype neurons. However, the contribution is most likely small (Figure 3 +0.6 Hz, Figure 5 +0.3 Hz). Again, the high variability of mEPSC frequency in the different experiments makes a direct comparison very difficult. Thus, my major concern is that experiments shown in Figure 3, Figure 5 and Figure 6 do not allow drawing conclusions about the relevance of cAMP-mediated increase in GluA3 function for synaptic plasticity in CA1 neurons. That is for example in contrast to the study of Gutierrez-Castellanos et al. (Neuron 2017) that convincingly showed that GluA3 channels and the novel GluA3 plasticity play a role for synaptic function of Purkinje cells (mEPSC frequency, amplitude, LTP) and mouse behaviour (VOR).

We observed that the variability between experiments is a consequence of differences in basal synaptic transmission between animals. In the revised Figure 3, we only compare control and forskolin condition between neurons of slices of the same age and obtained from the same animal. To allow us to compare between experiments, we calculate the ‘fold change’ (forskolin relative to control). These experiments demonstrate that the forskolin-mediated synaptic potentiation is in large part dependent on the presence of AMPAR subunit GluA3.

In the revised manuscript we added a new set of experiments (Figure 7) showing that β-adrenergic signaling in the hippocampus induces GluA3-dependent synaptic potentiation in CA1 neurons, thereby providing a more physiological form of this type of plasticity.

In all experiments throughout the revised manuscript we consistently show that GluA3 significantly contributes to the cAMP-mediated AMPAR potentiation. In Figure 3, Figure 5, Figure 6, and in the newly added Figure 7 and Figure 8 we show that a rise in cAMP (either through application of cAMP, FSK or β-ADR activation) in all cases significantly potentiates synaptic currents in a GluA3-dependent fashion. In addition, we added a new experiment in which we show that selectively knocking-out GluA3 in CA1 neurons in a conditional GluA3-KO model prevents synaptic potentiation upon NE-release in the hippocampus (Figure 8), indicating that the absence of a cAMP-mediated synaptic potentiation is not a consequence of aberrant neuronal developmental.

2) What was the rational to use acute brain slices from 3-5 weeks old mice in some experiments and organotypic slices of newborn mice in others? The mechanisms of FSK potentiation of synaptic currents might very well differ in the two preparations.

The rational for using acute brain slices in Figure 6 is that these experiments require whole cell recordings to be stable for up to 30 minutes, which in our hands are somewhat more efficient when performed in acute slices compared with organotypic slices.

We note that our findings in acute brain slices closely match those in organotypic slices. For instance, the application of β-ADR agonist isoproterenol gave a similar level of synaptic potentiation in acute versus organotypic slices (see Author response image 2).

**Author response image 2. respfig2:** Incubation of wild-type CA1 neurons with isoproterenol and IBMX increased the mEPSC frequency, but not amplitude.

3) The conclusion that Ras plays a role in GluA3-plasticity and competes with PKA-mediated GluA1-plasticity is not convincing. Except for the small increase in mEPSC amplitude, Ras-IgG have no influence on mEPSCs. In addition, experiments in Figure 6 are again very difficult to interpret since variability is high. Thus, is the lack of an effect of FSK explained because RAS-IgG increases mEPSC frequency in the -FSK control condition? mEPSC frequency in the presence of FSK is very similar irrespective of presence or absence of PKI or RAS-IgG (Figure 6). In addition, what was the rational to compare Fold-changes in frequency in Figure 6 instead of mEPCS frequency as in all other experiments? Finally, the number of neurons analyzed should be increased. In fact, I am surprised that the small difference in Fold-change in 6C reaches significance with an N of 5 and I am pretty sure that there would be no significant difference in the FSK-mediated increase in mEPSC frequency between IgG and RAS-IgG.

We have increased the N in our recordings from WT CA1 neurons, and we have repeated this experiment in GluA1-KO neurons to isolate the effect of Ras and PKA blockade on GluA2/3 currents. These new experiments revealed that the forskolin-driven increase in mEPSC frequency is significantly prevented when both Ras and PKA were blocked.

[Editors’ note: the author responses to the re-review follow.]

Reviewer #1:

This is a revised manuscript by Renner et al. showing that high concentrations of cAMP facilitates AMPA receptors through GluA3 subunits. The authors have performed several new experiments that clarify and support their previous conclusions. However, one of my previous concerns was the use of very high concentrations of cAMP or the use of FSK. To address this they tested whether activation of β-adrenergic receptors – a more physiologically relevant experiment – can also lead to GluA3 mediated plasticity. They indeed demonstrate that application of ISO evokes GluA3 potentiation, however only when PDEs are blocked. This suggests that the proposed GluA3 plasticity might take place under non-physiological conditions. They also show that i.p. injection of EPI in mice could lead to GluA3 AMPA receptor potentiation. This experiment is very difficult to interpret, as global activation of b-adrenergic receptors will initiate multiple downstream signaling pathways. Due to these reasons my enthusiasm for this work is still dampened.

We agree with the reviewer that GluA2/3-plasticity requires a robust increase in cAMP. Our finding that isoproterenol (ISO) by itself failed to induce GluA2/3-plasticity is actually not that surprising, considering that ISO is known generate a very weak rise in intracellular cAMP [Chay et al., 2016; Bruss et al., 2008]. This is because β-AR activation, besides adenylyl cyclase, also triggers the activation of PDEs, leading to a negative feedback loop, which significantly inhibits cAMP downstream signaling [Houslay and Baillie, 2005; Bruss et al., 2008]. This explains why we could only observe GluA2/3plasticity upon exposure of CA1 neurons to ISO when PDEs were inhibited.

The idea is that a robust increase in cAMP is only achieved when β-AR activation coincides with a rise in Ca^2+^ (e.g. upon NMDAR activation), which catalyzes cAMP production [Chetkovich and Sweatt, 1993; Wayman et al., 1994; Chay et al., 2016]. A short description of these considerations, including references, are added to the Results subsection “β-adrenergic signaling triggers the activation of GluA2/3-plasticity”.

We regret that this reviewer cannot bolster more enthusiasm for our experiments in which we show that β-AR activation in vivo upon Epi-injection is sufficient to trigger GluA2/3-plasticity. These data indicate that GluA2/3plasticity is active under physiological conditions.

*Reviewer #3:*

[…] I am still not convinced by the mEPSC data as the high variability of mEPSC frequency within one genotype precludes drawing conclusions about the contribution of GluA3 during cAMP mediated changes in synaptic function. A calculation of fold-change in frequency is not really helpful for an estimation of the contribution of GluA3. Absolute changes in frequency have to be compared for such an estimation. The very small increase in mEPSC frequency in GluA1 KO mice (approx. 0.1 Hz in Figure 6) compared to the much bigger increase in wt mice (approx. 0.5 Hz also in Figure 6) suggests that the contribution of GluA2/3 heteromers is rather small. Please discuss.

In our experiments an increase in mEPSC frequency is predominantly the result of the sizes of mEPSCs rising above the 5 pA detection limit. An increase in mEPSC frequency cannot be directly quantified into the absolute level of the change in postsynaptic strength. It can be compared between conditions as a relative change in synapse strength, provided that both conditions show a similar distribution of mEPSC events around the 5 pA detection limit.

However, the average mEPSC amplitude and frequency are substantially lower in GluA1-KO neurons compared with WT ones, and when we plot the distribution of mEPSC events recorded from WT and GluA1-KO neurons as shown in Figure 6, the distribution of mEPSC events before FSK/IBMX application is clearly different in GluA1-KO versus WT neurons (see Figure 6—figure supplement 2). Thus, the absolute increase in mEPSC frequency cannot be directly compared between WT and GluA1-KO neurons.

Due to the absence of GluA1, GluA1-KO neurons have experienced little activity-dependent AMPAR plasticity, and as a consequence the synapses have not matured as much as WT neurons and synapses contain on average lower amounts of AMPARs. In WT neurons GluA1-containing AMPARs at synapses are gradually replaced by GluA2/3s. For this reason, it is conceivable that WT neurons have more GluA2/3s at synapses compared with GluA1-KO neurons. This notion may well explain why the absolute change in mEPSC frequency upon the activation of GluA2/3-plasticity is larger in WT neurons compared with GluA1-KO neurons.

In conclusion, the larger increase in mEPSC frequency in WT neurons compared with GluA1-KO neurons does not reflect a small contribution of GluA2/3 plasticity to the cAMP-driven synaptic potentiation.

Results in experiments in Figure 1 and Figure 3 can be directly quantified, and these indicate that a rise in cAMP driven by FSK lead to a two-fold increase in both extrasynaptic (Figure 1) and synaptic (Figure 3) AMPAR currents. Importantly, FSK does not increase AMPAR currents in GluA3-KO neurons, indicating that the cAMP-driven increase in synaptic currents is predominantly a consequence of GluA2/3-plasticity.

To clarify this issue, we have included the distributions of mEPSC events as Figure 6—figure supplement 2 in the revised manuscript.